# Chromatin compaction during confined cell migration induces and reshapes nuclear condensates

Jessica Z. Zhao [1], Jing Xia[1] & Clifford P. Brangwynne [1,2,3,4] ✉

Cell migration through small constrictions during cancer metastasis requires significant deformation of the nucleus, with associated mechanical stress on the nuclear lamina and chromatin. However, how mechanical deformation impacts various subnuclear structures, including protein and nucleic acid-rich biomolecular condensates, is largely unknown. Here, we find that cell migration through confined spaces gives rise to mechanical deformations of the chromatin network, which cause embedded nuclear condensates, including nucleoli and nuclear speckles, to deform and coalesce. Chromatin deformations exhibit differential behavior in the advancing vs. trailing region of the nucleus, with the trailing half being more permissive for de novo condensate formation. We show that this results from increased chromatin heterogeneity, which gives rise to a shift in the binodal phase boundary. Taken together, our findings show how chromatin deformation impacts condensate assembly and properties, which can potentially contribute to cellular mechanosensing.

The cell nucleus is a complex, crowded environment that hosts many dynamic processes including transcription, RNA splicing, and DNA damage repair. Many of these processes are spatially compartmentalized around chromatin, the genetic material composed of nucleic acids and histone proteins, to carry out specific functions on the genome. Embedded on and around chromatin are a host of membrane-less structures known as biomolecular condensates, such as nucleoli, Cajal bodies, and nuclear speckles, which assemble through the physical mechanism of liquid-liquid phase separation (LLPS), and related phase transitions[1–8]. Consistent with their deformable fluid-like nature, biomolecular condensates have been shown to be sensitive to mechanical stresses: they can dynamically rearrange by coalescence under gravitational force, and undergo fission or even dissolve in the presence of pulling forces[9–11].

As the largest and relatively stiff organelle, the cell nucleus is a sensor of internal and external mechanical forces demonstrated in many physiological settings[12]. In polarized cells, nuclei respond to cytoskeletal force by asymmetric positioning[13,14]. When cells migrate in confined environments during development and cancer metastasis,

the nuclei undergo severe deformation to squeeze through tiny tissue spaces[15,16]. Two well-characterized major contributors to bulk nuclear mechanics are the lamina, localized on the periphery inside of the nucleus, and chromatin, forming a heterogeneous network distributed throughout the nuclear interior that can interact with peripheral lamina[17–21]. Changing the composition of the lamina filament network or the chromatin state can alter overall nuclei stiffness and viscoelastic properties[22–25]. Yet It remains unclear how condensates in the nucleus behave when cells are subjected to bulk external forces.

The possibility that chromatin deformation can impact the behavior of associated nuclear condensates is consistent with a number of findings on how condensates are closely coupled to the molecular crowding and mechanics of their local environment. At equilibrium, phase separation occurs when the concentration of molecules increases beyond their supersaturation level, $C_{sat}$. Previous studies show that osmotic stress, cell volume changes, or dynamic changes in protein concentrations can induce or suppress phase separation events in cells, by moving cells across the phase boundary[26–32]. Condensate size and form are also impacted by their

[1]Department of Chemical and Biological Engineering, Princeton University, Princeton, NJ, USA. [2]Princeton Materials Institute, Princeton University, Princeton, NJ, USA. [3]Howard Hughes Medical Institute, Princeton University, Princeton, NJ, USA. [4]Omenn-Darling Bioengineering Institute, Princeton University, Princeton, NJ, USA. ✉e-mail: cbrangwy@princeton.edu

mechanical surroundings. In the cytoplasm, condensates are impacted by the surrounding cytoskeleton[33], while nuclear condensates prefer to localize and grow in mechanically softer, low-density chromatin regions[34,35].

The close coupling between nuclear mechanics and condensate behavior is an emerging theme in mechanobiology[36]. Nuclear condensates are not only sensitive to their mechanical surroundings, but they can also reshape the chromatin through capillary forces[37,38]. Moreover, force transmission between condensates and their surroundings can be long-range (> 30 μm), as demonstrated in a recent study in mouse oocytes[39]. This raises the possibility that condensates may respond to forces generated during the highly mechano-active process of cell migration, in which cytoskeletal force generation and squeezing through constrictions can generate major internal stress and strain[40–42]. However, it is still largely unexplored how physical deformation during cell migration might impact the structure and phase behavior of nuclear condensates.

Here, we show that nuclear mechanical deformation during confined migration of human breast cancer cells impacts chromatin-embedded condensates. Using a microfluidics confined migration assay, we show that endogenous and synthetic condensates deform significantly due to stresses generated in the surrounding chromatin network. These stresses manifest differently in the advancing vs. trailing half of the constricted nucleus, leading to a differential chromatin distribution and giving rise to preferential phase separation in the trailing half. These results could suggest how new compartmentalization generated upon nucleus deformation potentially facilitates nuclear mechanosensing during confined migration.

## Results
### Mechanical deformation of the nucleus affects sub-nuclear structures during cancer cell migration in PDMS microfluidic device
To investigate the effect of mechanical deformation on phase separation inside the nucleus of living cells, we used cultured breast cancer MDA-MB-231 cells migrating in polydimethylsiloxane (PDMS) microfluidic devices, an established system for studying confined cell migration[40,43]. The channel of these microfluidic devices is 5 μm tall, consisting of two regions: the confined region with circular pillars separated by 2 μm wide spacings that mimic interstitial space (e.g., extracellular matrix pores), and the unconfined control region with 15 μm spacings in between (Fig. 1a). In the 2 μm region, the nucleus in the living cell experiences significant mechanical deformation, generated as the cell pulls itself through the constriction. Cells are seeded on the side without FBS and are allowed to adhere and migrate to the side supplemented with FBS, due to chemotaxis. When migrating through the 2 μm constrictions, cells are squeezed by the pillars, and their nuclei are significantly deformed. Studying confined migration using this system together with quantitative live-cell fluorescence microscopy allows measurements of nuclear deformation and condensate dynamics with high spatial-temporal resolution. Throughout the entire migration duration, cells show nuclear deformation in the 2 μm channel but not 15 μm, with visible cytoplasmic structures protruding through the channels (Fig. 1a, zoom-in). After migrating through PDMS pillars in both confined and control regions, cells are still able to undergo mitosis (Supplementary Fig. S1).

Using this system, we investigate the interplay between phase-separated nuclear bodies and nuclear mechanics under mechanical deformation throughout the confined-migration event (Fig. 1b). We hypothesize that mechanical deformation can reshape nuclear condensate dynamics due to their physical interactions with the surrounding chromatin environment, making them more prone to fuse or undergo fission as the cell nucleus mechanically squeezes through the narrow constriction (Fig. 1c). Mechanical deformation could also potentially result in a change in phase equilibrium in the cell nucleus,

through mechanochemical changes that promote or inhibit the formation of new nuclear condensates (Fig. 1c).

We chose nucleoli and nuclear speckles as two prominent and well-known examples of nuclear condensates. Using NPM1-mCherry as a marker for nucleoli and SRRM1-eYFP for nuclear speckles, we examined their dynamics throughout the confined migration process. During confined migration through the microfluidic device, we observed nucleoli protruding into the constrictions and elongating as the nucleus squeezes through, consistent with previous studies[19]. In some cases, nucleoli much larger than 2 μm in diameter, undergo fission events as they squeeze through the constriction, followed by the coalescence of smaller nucleoli to a bigger one nearby after the cell has migrated through (Fig. 1d). For both nucleoli and nuclear speckles, we observed fusion events occurring when the nuclei enter the constrictions (Fig. 1e, f). The coalesced nucleoli remain fused even after the cells exit the constrictions and the nucleus shape relaxes to an ellipsoidal shape. Further quantification shows that fission and fusion events are more frequently observed for cells migrating through 2 μm channels than control groups migrating in 15 μm channels (Fig. 1g–i), suggesting that nucleus deformation during confined migration may enhance nuclear condensate dynamics.

### Chromatin material response to nucleus deformation
Previous studies on condensate coarsening suggest that the local mechanical environment of the chromatin could influence condensate dynamics[6,35,44]. To quantify structural changes to chromatin and associated condensates across cells with various nucleus sizes and times of confined migration, we define the degree of progression through the constriction as a 'pseudo-time' parameter, calculated as $P = 1 - \frac{Area_{Trailing}}{Area_{Total}}$, where $Area_{Trailing}$ is the area of the trailing portion of the nucleus, and $Area_{Total}$ is total nuclear area, defined at each recorded frame (Fig. 2a). For example, for a symmetric nucleus at the midpoint of constriction where the advancing and trailing compartments are equal in size, $P = 0.5$. To quantify the chromatin density, we expressed H2B-mGFP as a marker for labeling bulk chromatin and quantified the mean H2B intensity in the nucleus in each frame and normalized it by the mean intensity just before the nucleus begins entering the constriction (when $P = 0$). As cells progress through the constrictions, the normalized mean intensity of chromatin marker H2B across entire nuclei does not change significantly, similar to the unconfined control group (Supplementary Fig. S2a). To quantify the change in the nuclear size, we measured the nuclear area marked by H2B, normalized by the average nuclear area at $P = 0$. We observed that there is no major decrease in average nucleus area as cell nuclei progress through constrictions (Supplementary Fig. S2b). To quantitatively assess chromatin reorganization during confined migration under mechanical force, we determined the chromatin flow field using particle-imaging velocimetry (PIV)[45,46]. The average chromatin displacement map shows that the velocity magnitude is the highest within the constriction, with coherent motion in the migration direction (Fig. 2b). This suggests that there are large local density changes during nuclear deformation.

To probe where chromatin spatially condenses locally, we imaged a cell line expressing both nucleolus granular component (GC) marker NPM1-mCherry and chromatin marker H2B-mGFP (Fig. 2c). It is known that chromatin frequently interacts with nucleoli through perinucleolar heterochromatin[47], and we wondered how this chromatin behaves during confined migration. We observed that perinucleolar H2B intensity increases as nucleoli deform. Interestingly, the intensity increases even after the majority of the cell passed through when $P = 0.7$ (Fig. 2c). This likely reflects nucleoli often lagging behind the constriction, while most of the nucleus has already progressed through, leading to compacted perinucleolar chromatin even at late stages of progression.

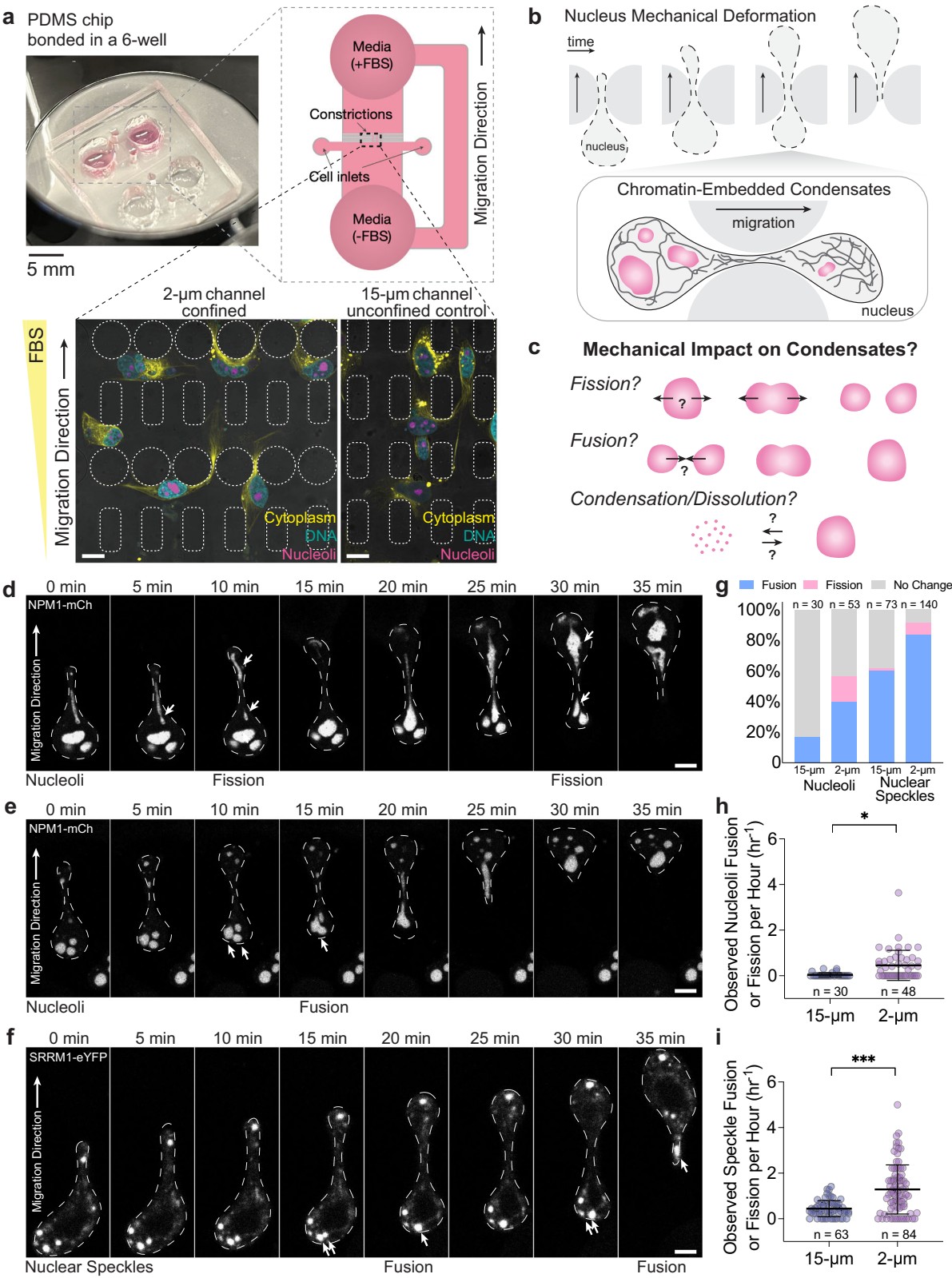

The difference in chromatin behavior in the advancing vs. trailing half of the cell nucleus suggests that the chromatin mechanical environment of the two compartments may be distinct. To compare the difference in H2B density in the two compartments, we quantified the heterogeneity by measuring the standard deviation of H2B pixel intensity divided by the mean H2B pixel intensity in the advancing or trailing half (Fig. 2d). The normalized heterogeneity of the trailing compartment shown by the magenta curve increases significantly as the nucleus progresses, while heterogeneity of the advancing compartment significantly decreases (Fig. 2d). Under conditions of increasing heterogeneity, these mechanically permissive regions are more prevalent, albeit adjacent to less mechanically permissive regions, such as perinucleolar heterochromatin. On the other hand, under more uniform chromatin conditions, there are no low-density

**Fig. 1 | Investigating the impact of mechanical deformation on phase-separated assembly using a microfluidics-based confined migration assay. a** top, Schematics of the imaging-compatible microfluidic device in a 6-well plate for studying cancer cell confined migration. **a** bottom, Human breast cancer cells MDA-MB-231 directionally migrate in the microfluidic device in the presence of an FBS gradient. Left: confined area with 2 μm wide channels where the cell nucleus undergoes mechanical deformation. Right: control area with a 15 μm wide channel where no major deformation of the cell nucleus is observed. Cyan: H2B-mGFP. Magenta: nucleoli marked by NPM1-mCherry. Yellow: cytoplasm marked by CellBrite dye. Scale bar: 15 μm. **b** Illustration of nuclear bodies in the nucleus undergoing mechanical deformation throughout the confined-migration event. Dashed line: outlines of the nucleus during confined migration. Zoom-in: various nuclear condensates (magenta) embedded in the chromatin environment (black line). **c** Schematics of different hypothetical outcomes of mechanical deformation impacting phase-separated assemblies inside the cell nucleus. Deformation can result in fusion and fission of nuclear condensates as the nucleus progresses

through constriction, or lead to nuclear protein condensation or dissolution. **d** Confined migration leads to nucleoli fission. Representative examples of NPM1-mCherry labeled nucleoli. Scale bar: 5 μm. **e** Representative examples of NPM1-mCherry labeled nucleoli fusing during confined migration. Arrows: fusion events. Scale bar: 5 μm. **f** Representative examples of SRRM1-eYFP labeled nuclear speckles fusing during confined migration. Arrows: fusion events. Scale bar: 5 μm. **g** Distribution of observed outcomes of nucleoli and nuclear speckles in a 150-minute time interval, with n indicating the number of nucleoli or nuclear speckles that can undergo fusion, fission, or no change in its dynamics. **h, i** Observed fusion and fission events per hour for nucleoli and nuclear speckles during confined migration. $n = 30$ cells expressing NPM1-mCherry in 15 μm channels, $n = 48$ cells expressing NPM1-mCherry in 2 μm channels, and $n = 63$ cells expressing SRRM1-eYFP in 15 μm channels, $n = 84$ cells expressing SRRM1-eYFP in 2 μm channels, across $N = 3$ independent experiments. Error bars show standard deviation, and statistical significance $*p = 0.04$, $***p < 0.001$ by one-way ANOVA for multiple comparisons (two-tailed).

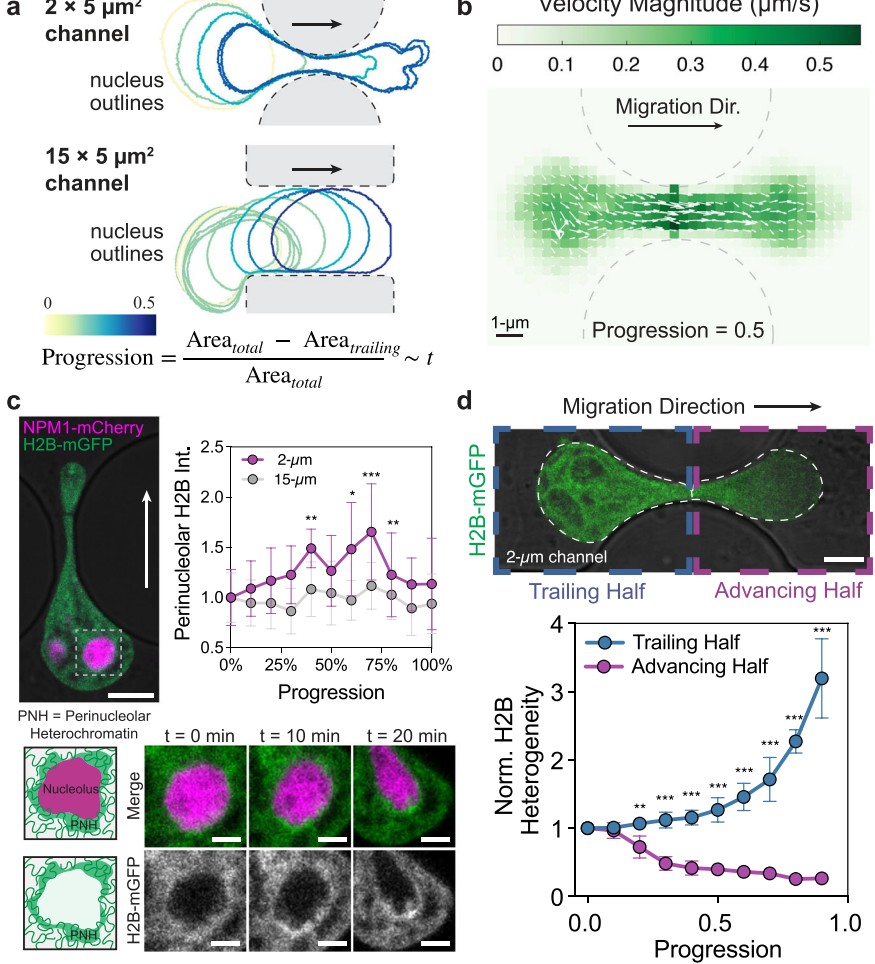

**Fig. 2 | Chromatin dynamics and distribution in response to nuclear deformation. a** Time lapse of nuclear outlines of representative cells migrating through confined or control channels. The colors of the nuclear outlines correspond to 'progression,' a parameter calculated based on the nuclear area at each time point, indicated in the equation shown. **b** Chromatin flow mapped by H2B-mGFP particle imaging velocimetry. Averaged velocity flow field of 19 nuclei. **c** Local compaction of perinucleolar heterochromatin in deformed nuclei undergoing confined migration. Magenta: NPM1-mCherry. Green: H2B-mGFP. The plot shows the average intensity of H2B surrounding the nucleoli over migration progression. $n = 23$ cells in 2 μm channels and 17 cells in 15-μm channels over $N = 3$ independent experiments. Error bars are standard deviations, and statistical significance $**p = 0.007$ (at

Progression = 0.4), $*p = 0.04$ (at Progression = 0.6), $***p < 0.001$ (at Progression = 0.7) and $**p = 0.006$ (at Progression = 0.8) by one-way ANOVA (two-tailed) comparing 2 μm vs 15 μm across multiple progression time points. Scale bars: 5 μm for upper image, 1 μm for lower cropped images. **d** Differences in chromatin compaction state in different regions of the nuclei during confined migration. The plot shows the normalized H2B heterogeneity within the trailing (blue curve) and advancing (purple curve) compartment over migration progression. Error bars are standard deviations of 16 cells across $N = 3$ independent experiments. $**p = 0.003$, $***p < 0.001$ according to one-way ANOVA (two-tailed) comparing advancing half vs trailing half at each progression time point. Scale bar: 5-μm.

chromatin regions in which the condensates can form. Combining this result with the constant overall H2B average intensity (Supplementary Fig. S2), this suggests that while the advancing and trailing halves have a similar amount of chromatin during confined migration, the deformation increases regions of low-density chromatin as well as nearby chromatin compaction in the trailing end.

### Condensates show distinct phase behavior in advancing vs. trailing half of the deformed nucleus

To probe the difference in chromatin mechanics across the two compartments, we used light-induced droplets as rheological probes, building from previous work[35]. We induce nucleation of synthetic Corelet condensates by illuminating cells in the microfluidic device with 488 nm blue light. Briefly, upon blue light activation, the 24 mer core scaffold consisting of iLID-GFP-FTH1 binds to FUS$_N$-mCherry-sspB through iLID-sspB interactions and forms condensates via interactions between the FUS$_N$ intrinsically-disordered region (IDR) (Fig. 3a). To reduce heterogeneity arising from varied nucleus shapes, we form Corelet droplets when the nucleus has squeezed halfway into the constriction, by illuminating the nucleus with 488 nm light for 3 min.

Interestingly, we noticed that droplets nucleated in the trailing end of the nucleus tend to be larger, compared to ones localizing at the advancing end, and the middle portion within the 2 μm channel is densely packed with chromatin (Fig. 3b). This suggests that the increased heterogeneity in the trailing half of the nucleus gives rise to effectively softer local mechanics, where droplets preferentially form due to the more mechanically permissive environment. To further test this idea, we track the fluctuating motion of these droplets over time and calculate pairwise mean-squared displaces of droplets tracked at a fast frame rate, to avoid the confounding influence of bulk translational motion. Despite their larger average sizes, we find that the mean-squared displacement of droplets in the trailing half is larger than that in the advancing half (Fig. 3c), consistent with a softer mechanical environment. Indeed, measurements of probe particle mobility provide further support for differences in the mechanical properties in the advancing vs. trailing half, with interesting dependence on the experimentally accessible length and time scales (Fig. S3, Supplementary Discussion).

Given the apparently more heterogeneous and mechanically soft environment at the trailing half of the cell nucleus, we hypothesized that it could also be more permissive to the initial nucleation of newly forming condensates. Consistent with this prediction of preferential nucleation in the trailing half, we observe that in the cells expressing the Corelets components close to the phase boundary, condensates nucleate only in the trailing half of the nucleus, but never observe the opposite (Fig. 3d and Supplementary Fig. S5). By measuring the local concentrations of the core and sspB in the advancing vs. trailing of the nucleus, we mapped their locations on the Corelet phase diagram. By taking the ratio of core to IDR-sspB, we can measure the inverse of valence on the y-axis, valence$^{-1}$, where valence is the multiplicity of IDR decoration of the 24 mer cores, and reflects their ability to participate in multivalent interaction with one another. Results showed that the points quantified for the trailing half are located deep in the 2-phase regime, while the points for the advancing half are located very close to the binodal phase boundary (Supplementary Fig. S5).

To gain further insight into the nature of this preferential condensate nucleation we observe with Corelets, we sought to determine if it can also occur with non-optogenetically activated condensates. We selected 53BP1 as a candidate protein due to its ability to condense through phase separation and the high self-association propensity through oligomerization and long stretches of IDRs[48]. We generated monoclonal MDA-MB-231 cell lines expressing miRFP670-53BP1 and H2B-mGFP and screened for clones with low 53BP1 expression levels, such that they do not phase separate spontaneously before squeezing through confined spaces. In these cells, we observe 53BP1 forms de novo condensates as the nucleus squeezes through the 2 μm constrictions (Fig. 3e). These condensates dissolve after nuclei restore their shape when they exit out of the constrictions (Fig. 3e, "After"). Moreover, we find that these mechanically-induced 53BP1 condensates preferentially localize in chromatin-poor regions, consistent with our hypothesis that the trailing compartment is mechanically softer and thus more permissive to phase separation (Fig. 3e, zoom-in).

One potential explanation for our findings with induced 53BP1 condensates is that these are forming due to DNA damage since 53BP1 localized to DNA damage repair foci in the presence of double-stranded breaks (DSBs), which can occur due to nuclear envelope rupture in such a microfluidic cell migration system[49–51]. To examine this possibility, we performed siRNA knockdown of RNF168 (Supplementary Fig. S6a, b), a double-stranded break repair protein upstream of 53BP1, and repeated the cell migration assay in the same experimental conditions. We observe the same phenomenon as that in the untreated cell line (Supplementary Fig. S6c). Moreover, by overexpressing a full-length tudor-domain mutant that abolishes histone-binding function in DSBs repair, we still observe the formation of de novo 53BP1 condensates (Supplementary Fig. S6d). Furthermore, by expressing mCherry alone with a nuclear localization sequence (NLS-mCherry), we can monitor nuclear envelope rupture events by the presence of mCherry signal in the cytoplasm. We find that 53BP1 de novo condensate formation can happen without nuclear envelope rupture (Supplementary Fig. S6e). Thus, the formation of 53BP1 condensates is not simply a consequence of DSB induced by confined migration.

To further examine the generality of the findings of de novo and preferential condensation in the trailing half of constricted nuclei, we also tested BRD4, a protein well-known to be capable of phase separation, mediated through a large IDR on the C-terminus[34,52,53]. Similar to 53BP1, we observe de novo BRD4 condensates forming upon mechanical deformation in confined cell migration, with preferential condensation in the chromatin-poor region, consistent with previous literature[34] (Fig. 3f, zoom-in). For both 53BP1 and BRD4 constructs, condensation behavior is reversible and favors the trailing half of the nucleus during confined migration (Fig. 3g–j). To rule out the possibility that serum level affects protein condensation behavior, we quantified the number of condensates in cells grown in media containing no serum versus cells grown in 10% FBS-supplemented media, and observed no difference in morphology or number of condensates (Supplementary Fig. S7). Taken together, these data suggest that at least three different types of proteins, tuned both optogenetically, and through overexpression close to the saturation concentration, exhibit induced condensation preferentially in the trailing end of the nucleus during confined migration.

### Local chromatin mechanics impact condensate formation

We reasoned that the increased heterogeneity in the trailing half of the nucleus would give rise to locally soft chromatin regions, which might shift the phase boundary to enable more permissive condensation. To test this idea in an experimental paradigm orthogonal to our confined migration setup, we sought to control the degree of chromatin compaction in unconfined cells using chemical perturbation methods that are known to cause global chromatin reorganization. We examined the consequences of chromatin compaction induced by hyperosmolarity via 5% sorbitol solution, which gives rise to more highly segregated miRFP670-H2B labeled chromatin, quantified using a normalized heterogeneity metric (Fig. 4a and Supplementary Fig. S8). Conversely, we can achieve more uniform chromatin through decompaction, achieved by decreasing total heterochromatin by using a broad-spectrum histone methyltransferase inhibitor (HMTi) 3-Deazaneplanocin-A (DZNep)[23] (Fig. 4a).

To test whether these changes in chromatin heterogeneity can lead to changes in condensate phase equilibria (Fig. 4b), we expressed

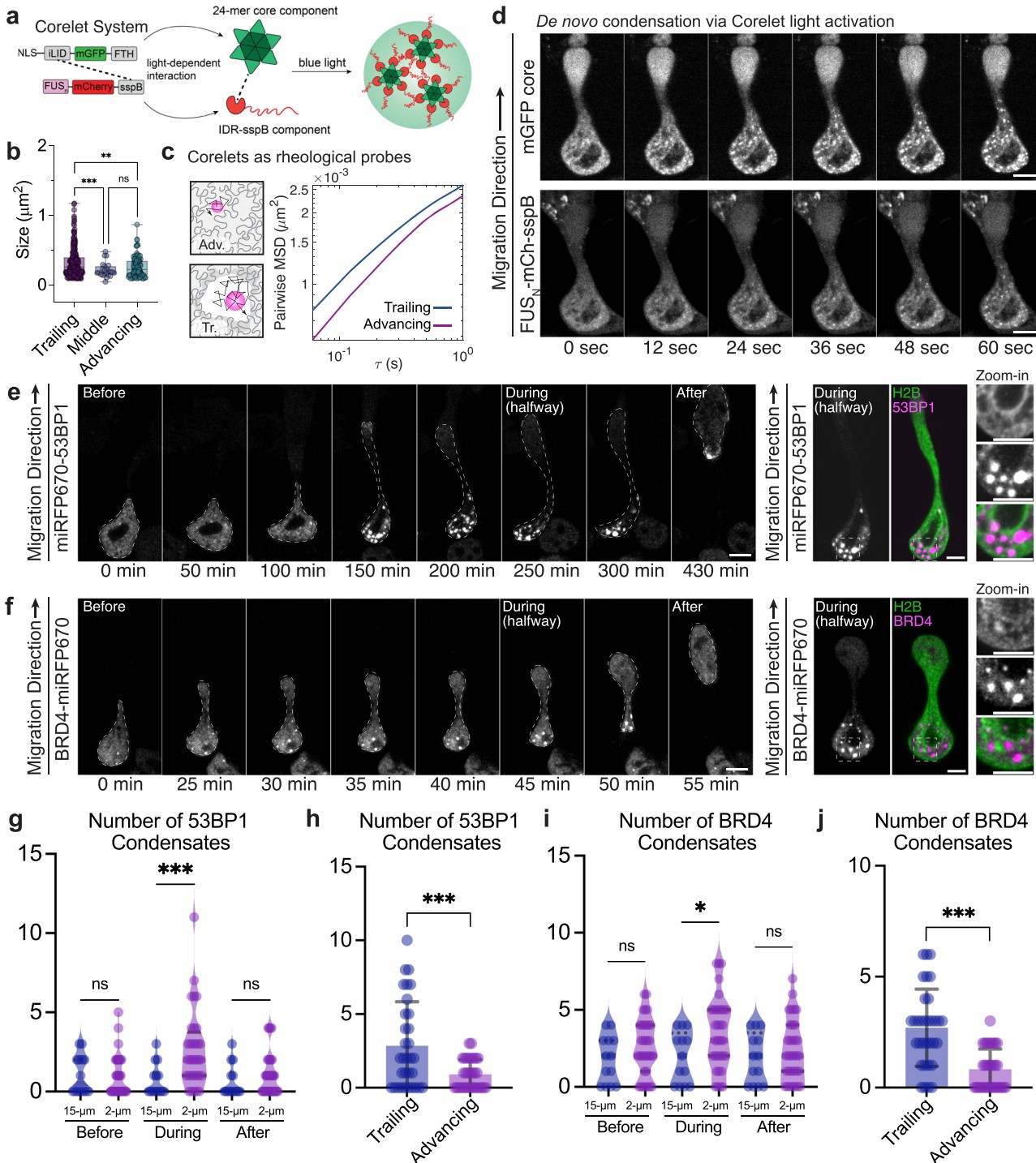

FUS$_N$ Corelets components in cells seeded in regular well-plates (i.e., unconfined cells). We quantitatively constructed the Corelet phase diagram to determine binodal phase boundaries of Corelets, as previously described[54]. Consistent with our hypothesis, the DZNep-treated population, with decreased chromatin heterogeneity, exhibited a downward shift in the binodal boundary, implying suppressed phase separation (Fig. 4c). Conversely, cells treated with 5% sorbitol, and increased chromatin heterogeneity, exhibited an upshift of the binodal boundary, implying enhanced phase separation (Fig. 4d). To further test whether this shift in the binodal phase boundary affects condensate behaviors, we treated Corelet-expressing cells seeded for confined migration with 5 μM DZNep, and characterized Corelet size after 3 min light activation (Supplementary Fig. S9A). Quantification

shows no significant difference in Corelet size across the advancing versus trailing half (Supplementary Fig. S9b). Thus, chromatin heterogeneity can tune the phase equilibrium inside the living cell nucleus, likely underlying our observations of confined migration-induced condensation in the trailing half of the nucleus.

## The concentration of interchromatin space-occupying proteins increases in the trailing half of migrating cells

Another potential contributing factor to our observations of migration-induced condensation is changes in protein concentration, which could potentially shift sub-regions of the cell across the phase boundary (Fig. 5a). We quantified the mean fluorescent intensity of protein in the trailing half, normalized by mean fluorescent intensity across the whole

**Fig. 3 | Condensate-forming proteins show distinct phase behavior in advancing vs. trailing half of the deformed nucleus. a** Schematics of Corelet activation experiment to test whether Corelet droplets prefer to nucleate in the trailing half. **b** Scatterplot shows the size difference of corelet droplets post 3-min activation in trailing, middle (a narrow region in 2 μm channel), and advancing halves. **\*\*p** = 0.002578 and **\*\*\*p** < 0.001 according to the one-way ANOVA test (two-tailed). Error bars are standard deviations of $n$ = 244, 23, and 109 Corelet condensates in $n$ = 14 cells for trailing, middle, and advancing groups, respectively, in $N$ = 3 independent replicates. **c** Chromatin mechanics probed by light-induced corelets shows the difference in advancing vs. trailing half in $n$ = 16 cells at $P$ = 0.5, in $N$ = 3 independent replicates. Pairwise MSD shows a difference in the mobility of droplet populations in the two halves. **d** Representative cell with de novo Corelet droplets forming only in the trailing half. Scale bar: 5 μm. **e** De novo formation of 53BP1 condensates as the nucleus migrates through confinement. Condensates localize at

chromatin-poor regions (right). Magenta: miRFP670-53BP1, green: H2B-mGFP. Scale bar: 5 μm. **f** De novo formation of BRD4 condensates as the nucleus migrates through confinement. Condensates localize at chromatin-poor regions (right). Magenta: BRD4-miRFP670, green: H2B-mGFP. Scale bar: 5 μm. **g** Condensation of 53BP1 is reversible. **\*\*\*p** < 0.001 according to one-way ANOVA (two-tailed). $n$ = 33 nuclei (2 μm) and $n$ = 14 nuclei (15 μm) in $N$ = 3 independent experiments. **h** Condensation of 53BP1 biases towards the trailing half of the deformed nucleus. **\*\*\*p** < 0.001 by paired $t$ test (two-tailed). The error bar is the standard deviation of $n$ = 33 nuclei in $N$ = 3 independent experiments. **i** Condensation of BRD4 is reversible. **\*p** = 0.05 according to the one-way ANOVA test (two-tailed). $n$ = 32 nuclei (2 μm) and $n$ = 13 nuclei (15 m) in $N$ = 3 independent experiments. j Condensation of BRD4 biases towards the trailing compartment of the deformed nucleus. **\*\*\* p** < 0.001 according to $t$ test (two-tailed). The error bar is the standard deviation of $n$ = 33 nuclei in $N$ = 3 independent experiments.

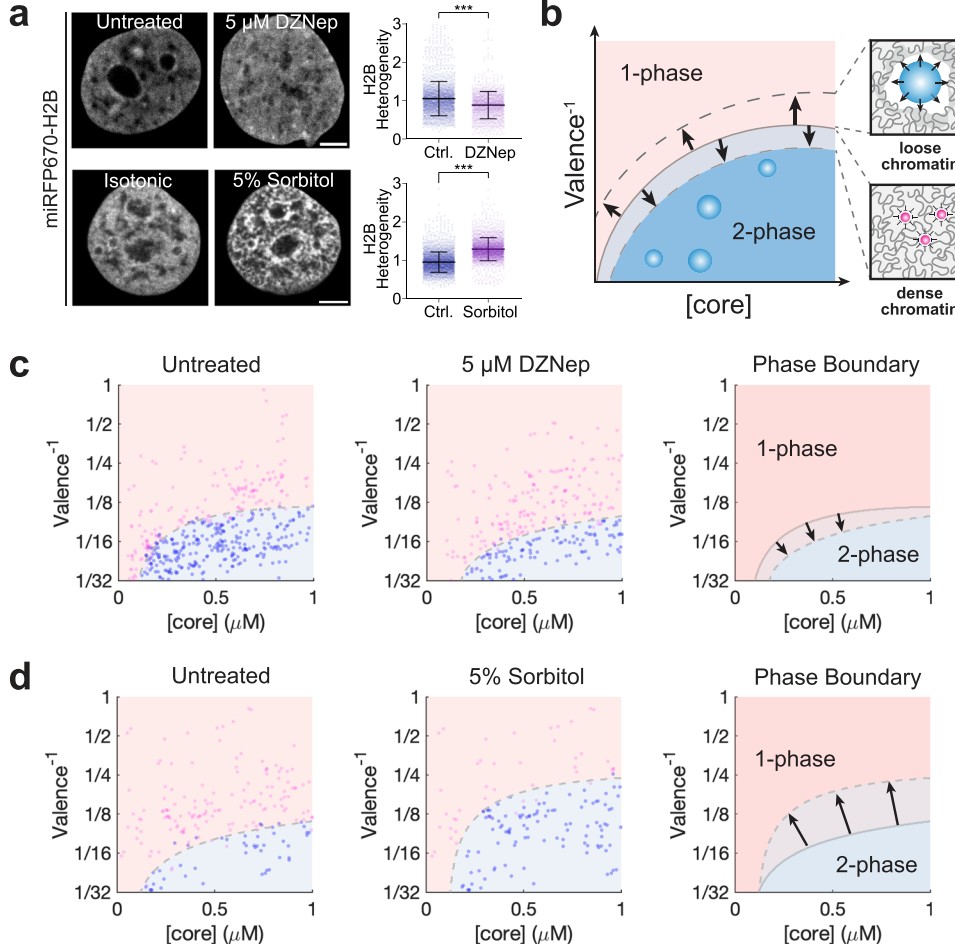

**Fig. 4 | Chromatin heterogeneity tunes nuclear condensates phase separation equilibrium. a** Representative image of cells expressing miRFP670-H2B under different conditions. Top: HMTi experiment; left: untreated; right: 5 μm DZNep treated for 24 h. Bottom: sorbitol compression experiment; left: pre-compression; right: 5% sorbitol compression for 30 min. Scatterplots show the difference in chromatin heterogeneity between different treatments. **\*\*\*p** < 0.001 according to a one-way ANOVA test (two-tailed) from 4 independent technical replicates for each condition. Error bars show a standard deviation of $n$ = 2191, 1247, 2720, and 2569 cells in untreated, DZNep, isotonic, and 5% sorbitol conditions, respectively. Scale

bar: 5 μm. **b** Schematics of phase diagram change resulted from the change in chromatin heterogeneity. When chromatin compacts, droplets are more prone to form due to increased interchromatin space. However, for decompaction, chromatin material in the densely packed region is released, increasing the overall chromatin density, and suppressing phase separation. **c, d** Phase boundary measured by Corelet activation under (**c**) 5 μM DZNep for chromatin decompaction or (**d**) 5% sorbitol for chromatin compaction. Each dot represents a single cell (2 min activation) that either is not phase-separated (magenta) or is phase-separated (blue). Best-fit phase threshold is shown. $n$ > 100 cells for each experiment.

nucleus, and plotted against nucleus progression. If proteins localize disproportionately in the trailing half, this quantity will increase above 1 as the cell progresses through the constriction (Fig. 5b). By quantifying this in all the cells completing confined migration, we show that for 53BP1, the relative average intensity has a statistically significant increase when cells reach $P$ = 0.5 (Fig. 5c). In cells expressing BRD4, relative

average intensity also increases significantly as well at $P$ = 0.5 (Fig. 5d). To further test this concentrating hypothesis systematically, we overexpressed various eYFP-tagged IDR-containing full-length proteins, including RNPS1, SART1 and SRRM1. For the eYFP tag alone, the relative intensity in the trailing half does not increase, suggesting that the addition of a disordered protein sequence is needed for retention in the

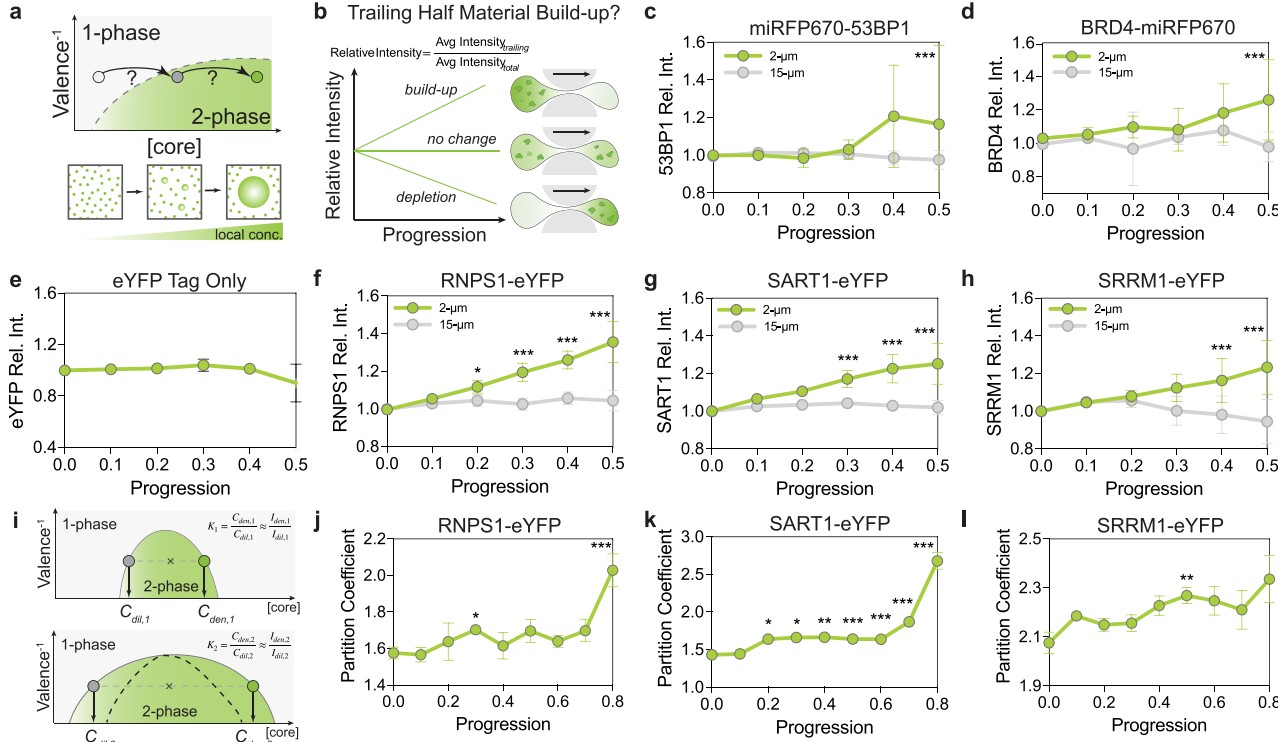

**Fig. 5 | Effect of mechanical deformation on local protein concentration.**
**a** Schematics showing how concentrating molecules can induce phase separation by pushing across the phase boundary. **b** Schematics of quantifying the effect of concentration increase. Relative average intensity measures how much material is disproportionately distributed in the trailing compartment. **c** Increase in trailing-end 53BP1 intensity as cells squeeze through constrictions. Error bars are standard deviations of 14 cells for 2-μm and 12 cells for 15-μm. ***$p < 0.001$ by one-way ANOVA test (two-tailed). **d** Increase in trailing-end BRD4 intensity as cells squeeze through constrictions. Error bars are standard deviations of 33 cells and 10 cells for 15-μm. ***$p < 0.001$ by one-way ANOVA test (two-tailed). **e** eYFP tag alone does not show material built-up associated with mechanical deformation. All progression bins have non-significant differences compared to Progression = 0. The error bar is the standard deviation of $n = 4$ cells. **f**–**h** Testing various constructs known to phase-separate in the cell nucleus: RNPS1-eYFP, SART1-eYFP, and SRRM1-eYFP. Error bars

are SD of 17, 6, and 27 cells for RNPS1-eYFP, SART1-eYFP, and SRRM1-eYFP, respectively, for the 2-μm group, and 10, 8, 14 cells for RNPS1-eYFP, SART1-eYFP, and SRRM1-eYFP, respectively, for 15-μm group. Insets show the change in the averaged partition coefficient of pre-formed condensates during migration progression. RNPS1-eYFP *$p = 0.01$, all subsequent ***$p < 0.001$ by one-way ANOVA test (two-tailed). Statistical analysis was done at each progression time point comparing 2-μm vs 15-μm channel groups. **i** Schematics showing how a shift in phase boundary manifests in an increase in partition coefficient, the ratio of $C_{den}$ to $C_{dil}$. **j**–**l** Measured partition coefficient of RNPS1-eYFP, SART1-eYFP, and SRRM1-eYFP. Error bars are SEM of 17, 7, and 27 cells for RNPS1-eYFP, SART1-eYFP, and SRRM1-eYFP, respectively. *$p = 0.04$ (RNPS1 $P = 0.3$), ***$p < 0.001$ (RNPS1 $P = 0.8$), *$p = 0.02$ (SART1 $P = 0.2$), *$p = 0.02$ (SART1 $P = 0.3$), **$p = 0.003$ (SART1 $P = 0.4$), ***$p < 0.001$ (SART1 $P = 0.5, 0.6, 0.7, 0.8$), **$p = 0.008$ (SRRM1 $P = 0.5$) by one-way ANOVA test compared to $P = 0$ (two-tailed).

trailing half (Fig. 5e). For RNPS1-eYFP, SART1-eYFP, and SRRM1-eYFP when expressed at a level above their saturation concentrations, these proteins are capable of forming chromatin-excluding condensates in the nucleus[34]. For the set of eYFP-tagged nuclear proteins tested, in all cases, the mean relative intensity increases significantly as cells squeeze through the constriction (Fig. 5f–h).

To connect to our earlier hypothesis that chromatin heterogeneity modulates the phase boundary shift, we note that a shift of the binodal line will result in a lower concentration of the dilute phase, and potentially a higher concentration of the dense phase (Fig. 5i). This phase boundary shift manifests as an increase in the partition coefficient, which is the ratio of the protein concentration inside to outside the condensate. To test this, we measure the partition coefficient of eYFP-tagged constructs, RNPS1, SART1, and SRRM1. As cells progress through the 2-μm constriction, the partition coefficient increases (Fig. 5j–l), in agreement with our prediction. Taken together, these results suggest that both an increase in concentration and a shift in the phase boundary can contribute to enhanced condensation in the trailing half during confined migration.

## Discussion
Biomolecular condensates are dynamic intracellular structures that can condense, dissolve, and rearrange during various cell processes,

and in response to biochemical perturbations such as altered ionic strength, crowding, or osmotic shock[26,55]. However, how condensates behave under direct mechanical force is less well-understood, especially within living cells. Our findings suggest that mechanical deformation of the cell nucleus can impact the formation of biomolecular condensates, supported by the following: (1) nucleoli and nuclear speckles, localized in close proximity to chromatin, undergo fusion and fission when cell nuclei squeeze through small constrictions; (2) within the same nucleus undergoing confined migration, chromatin in the advancing half loses heterogeneity while the trailing half increases heterogeneity and creates more open regions; (3) a variety of model condensates preferentially form de novo in the trailing half of the nucleus; (4) deformation-induced chromatin heterogeneity and altered protein concentration both appear to tune phase equilibrium. Taken together, our work shows that mechanical deformation of the cell nucleus can strongly impact the thermodynamics phase behavior of nuclear condensates (Fig. 6).

Cells regulate chromatin spatial organization and mechanics in response to mechanochemical cues in different tissue microenvironments[56]. It has been suggested that increased mobility and metastatic potential are associated with increased chromatin compaction, specifically through the increase in histone modifications H3K9me[3] and H3K27me[3] [57–61]. Chromatin decompaction through

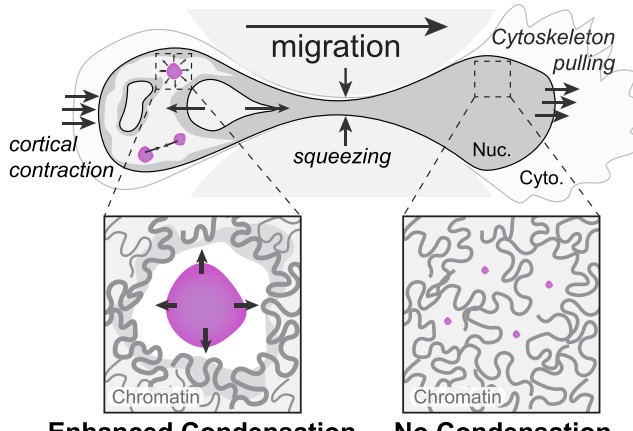

**Fig. 6 | A biophysical model for mechanochemical impact on form and assembly of nuclear condensates.** The cell nucleus experiences forces from various sources, such as pulling force from the cytoskeleton, cortical contraction force, and local compressive force from the confinement. Condensates respond to these forces by undergoing fusion and fission events and manifest different phase behaviors in the advancing vs trailing half of the nucleus during confinement.

pharmacological inhibition or overexpression of nucleosome-binding protein HMGN5 leads to softer nuclei and more frequent nuclear blebbing[58]. Our results show that the advancing vs. trailing halves of confined nuclei have different chromatin structural and mechanical properties: the advancing half is more homogeneously distributed while the trailing is more heterogeneous, giving rise to the distinct phase behavior for nuclear proteins across the two compartments. However, given the technical challenges of directly measuring the local mechanics of the nucleus within the confined microfluidics channels, we characterized chromatin using passive tracking of nanoprobes (light-induced Corelets and GEMs) and FRAP experiments in the nucleus. These approaches primarily reflect the local subcellular structure of the nucleus, or 'mesh size', which is linked to but not a direct measurement of the local mechanics. Despite the limitations, our results suggest that in the trailing half, a locally more open chromatin meshwork is energetically more favorable for condensate nucleation and growth.

Our findings on the role of chromatin heterogeneity during confined migration in modulating intracellular phase behavior are interesting to consider in light of findings in non-living systems showing that mechanical constraints from an elastic matrix can arrest phase separation of embedded molecules[62,63]. Condensates forming in the trailing end could thus suggest that chromatin mechanics in the trailing half is effectively 'softer'. However, ultimately the impact of chromatin on condensate phase behavior is a convolution of both structural and mechanical effects; the increased structural heterogeneity at the rear appears to give rise to a particularly phase-separation permissive local environment. Indeed, chromatin compaction as the nucleus is squeezed through the channel could redistribute the chromatin network, generating locally open and mechanically softer regions[64].

In our system, we showed that for nuclear proteins expressed at a level near their saturation concentrations, new condensates can form in response to mechanical stress. Large, pre-existing native condensates, including nucleoli and nuclear speckles, do not exhibit noticeably enhanced condensation or dissolution, likely due to being significantly above their supersaturation concentrations. However, these structures do deform significantly and can undergo fission and fusion events, highlighting their intimate mechanical coupling to the surrounding chromatin environment.

Interestingly, a recent study using a different microfluidic device geometry showed that increased expression of long non-coding RNA NEAT1 in MDA-MB-231 cells can promote the speed of the nucleus during confined migration[65]. The authors observed de novo formation of NEAT1 condensates in the advancing half upon selectively boosting NEAT1 isoform 1-2. This difference from our results might be due to the proportion of cell nucleus that is occupying the constriction geometry, which in that case is a long straight microchannel, leading to a difference in the direction of integrin-mediated protrusion force transmitted to the nucleus[66].

Could a more phase separation-permissive environment in the trailing end of a migrating cell nucleus contribute to cell function or dysfunction? Perhaps forming condensates in the trailing half promotes migration by mechanically supporting cells through confinement. Moreover, surfaces of nuclear condensates can create new self-interactions between chromatin loci or chromatin-protein interactions[37,38]. Our results showing the shift in local phase equilibrium might play a functional role in nucleating condensates as mechanical entities that create new chromatin interaction. This is consistent with previous reports that cells exhibiting high cancer metastatic progression have altered chromatin interactions in 3D and translocation of specific genomic elements[67–69]. Nevertheless, it is still an open question how nuclear condensate number and behavior correlate with metastatic potential in different cell types. It is possible that nuclear condensates formed upon mechanical deformation can also serve as a reaction crucible for the localization of mechanosensitive transcription factors, many of which are shown to be capable of phase separation in cells[27].

Our work reveals how mechanical triggers during confined migration are able to dynamically rearrange existing nuclear condensates and even give rise to de novo condensates which emerge from a shift in phase equilibrium due to altered chromatin mechanics. Our biophysical model provides an exciting opportunity for future work to investigate phase separation as a global mechanosensing mechanism in cells under mechanical force in various physiological and disease conditions. Future experiments combining confined migration experiments with microfluidics-based single-cell genomics assay are needed to interrogate the role of condensation on mechanosensitive gene expression and broader function.

## Methods

### Cell culture

All cell lines were maintained at 37 °C with 5% $CO_2$. The breast adenocarcinoma cell line MDA-MB-231 was a kind gift from Jan Lammerding. MDA-MB-231 cells and Lenti-X 293 T cells (Takara Bio #632180) were cultured in DMEM (GIBCO #11995065) with 10% (v/v) FBS (Atlanta Biological #S11150H) and 1% (v/v) PenStrep (GIBCO #15140122).

### Plasmid construction

All cell lines were transduced with lentivirus stably expressing one or combination of the following lentiviral plasmids for fluorescence microscopy: pHR-H2B-mGFP[35], pHR-NPM1-mCherry[70], FM5-SRRM1-eYFP[71], pHR-NLS-iLID-mGFP-FTH1[54], pHR-FUS$_N$-mCherry-sspB[54], FM5-miRFP670-53BP1[71], FM5-miRFP670-53BP1$^{D1521R}$ (this work), FM5-NLS-GEM40-mGFP (this work, cloned from GEM40 nanoparticle[32]), FM5-RNPS1-eYFP[34], FM5-SART1-eYFP[34], FM5-NLS-mCherry are kind gifts from David W. Sanders[72]. For constructing lentiviral plasmids, DNA fragments were amplified by PCR using CloneAmp HiFi PCR Premix (Takara Bio #639298) and the following PCR primers with D1521R point mutation included in the primer sequence:

FM5-miRFP-53BP1 D1521 1 F
5'-GTGGTAGCGGCAGTGGCGCGgaccctactggaagtcagttg-3'
FM5-miRFP-53BP1 D1521 R 1R

5′-acatcacattcgtacccGCGatcaaagagcaatttatac-3′
FM5-miRFP-53BP1 D1521R 2 F
5′-gctctttgatCGCgggtacgaatgtgatgtgttgg-3′
FM5-miRFP-53BP1 D1521R 2 R5′-GATAAGCTTGATATCGAATTttagt
gagaaacataatcgtgtttatattttggatg-3′

PCR reactions were then gel purified using NucleoSpin Gel and PCR Clean-Up kit (Takara Bio #740609). Purified PCR fragments were inserted into either linearized FM5 or linearized pHR lentiviral vector and assembled using an In-Fusion HD cloning kit (Takara Bio #638910). All constructs were sequence confirmed by Sanger sequencing.

## Generation of stable cell lines

Lentiviruses were used to generate MDA-MB-231 cell lines stably expressing fluorescent proteins. Lentiviruses were produced by plating Lenti-X 293 T cells (Takara Bio #632180) in 6-well plates at a confluency of 70–80% at the time of transfection. Lentivirus packaging plasmids psPAX2 (Addgene #12260) and pMD2.G (Addgene #12259) and lentiviral transfer plasmid were transfected into Lenti-X cells using FuGENE HD Transfection Reagent (Promega #E2311) incubated in Opti-MEM (GIBCO #31985062). After 48 h, lentivirus supernatants from the 6-well plate were harvested and filtered through a 0.45 μm filter. Lentivirus supernatant was further concentrated 10- to 20-fold using Lenti-X Concentrator (Takara Bio #631232), and immediately used or stored in aliquots at −80 °C. MDA-MB-231 cells were plated at 20–30% confluency and reverse-transduced with lentivirus. Viral media was changed to fresh media 24 h post-transduction, and 2-3 days post-transduction, cells were harvested for FACS sorting enrichment for the fluorescently labeled polyclonal population. To generate monoclonal MDA-MB-231 cell lines expressing H2B-mGFP and miRFP670-53BP1, single cells were sorted into individual wells of 96-well plates, and monitored for 2-3 weeks for colony growth, with filtered conditioned media replenished every 5–7 days. Clones #14 and #18 were selected for diffuse distribution of exogenously expressed miRFP670-53BP1 in the nucleoplasm without pre-existing puncta, and were used to conduct confined migration experiments.

## siRNA-mediated knockdown of RNF168

siRNA oligonucleotide was purchased from ThermoFisher, for knockdown of human *RNF168* (Silencer Select #s533834), with a negative control siRNA (Silencer Select #4404021). MDA-MB-231 cells were plated in 12-well plates at 30–50% confluency at the time of treatment, and reverse transfected with 25 nM siRNA diluted in Opti-MEM using Lipofectamine RNAiMAX Reagent (Invitrogen #13778030). Cells were collected 48-96 hours post-siRNA transfection for validation of knockdown using bleomycin assay in 96-well plates or seeding into the confined migration device.

## Immunofluorescence staining

Rabbit anti-53BP1 was purchased from Novus Biologicals (#NB100-305) for immunofluorescence staining of endogenous 53BP1 in MDA-MB-231 cells. Cells cultured in glass-bottom 96-well plates were fixed with 4% paraformaldehyde (Electron Microscopy Science #15710) in 1x PBS for 15 min at room temperature and washed 2 times with 1x PBS on a rocker for thorough rinsing. The samples were then permeabilized using 1x TBS-T containing 0.5% Triton X−100 for 15 min at room temperature on a rocker. Subsequently, the samples were blocked using 5% goat serum in 1x TBS-T containing 0.1% Triton X-100, and incubated overnight at 4 °C with the primary antibody against 53BP1 (rabbit anti-53BP1, dilution 1:50 in block buffer). Samples were then rinsed with 1x TBS-T 4 times with 5 min incubation in between on a rocker, and incubated with goat anti-rabbit AlexaFluor-647 secondary antibody (Invitrogen #A-21246). Finally, samples were washed 4 times with 1x TBS-T with 5 min incubation in between on a rocker, and mounted with vectashield antifade mounting medium (Vector Laboratories #H-1000).

## Western blot

To validate the knockdown of RNF168 via siRNA, we performed a western blot against RNF168 with three independently generated groups for negative control siRNA and siRNF168. Cell pellets from each group were resuspended and incubated in RIPA buffer (Thermo Scientific #89901) containing 100x Halt™ protease and phosphatase inhibitor cocktail (Thermo Scientific #78440) and 300x benzonase nuclease (Millipore Sigma #E8263) 30 min on ice. After BCA protein quantification (Pierce), 10 μg samples were subjected to denaturation with LDS sample reducing agent (Invitrogen #NP0004) and sample buffer (Invitrogen #NP0007). Anti-RNF168 primary antibody (Millipore Sigma #ABE367) was used at 1:1000, and secondary antibody (Jackson #111-035-144) was diluted at 1:10000. Loading control was stained with β-Tubulin antibody (Cell Signaling Technology #2146) at 1:1000 with the same secondary antibody at 1:10000.

## Osmotic compression assay

Sorbitol powder was dissolved into fresh, complete cell culture media at 5% w/w. Cells are seeded onto a 96-well plate and loaded onto the microscope. Sorbitol solution was added to cells, and the sample was imaged after a 30 minute incubation period to allow for thermal equilibrium. Cells were subsequently imaged using the high-throughput phase mapping protocol before and after adding sorbitol.

## Biochemical perturbation

To validate the knockdown of RNF168 via siRNA, bleomycin (Sigma #203408-10MG-M) was used as a DNA damage agent. The powder was dissolved in PBS at 10 mg/ml as stock concentration. A final concentration of 10 μg/ml was used to induce a double-stranded break in MDA-MB-231 cells. For the chromatin decompaction experiment, histone methyltransferase inhibitor 3-deazaneplanocin A (DZNep) (Cayman Chemical #13828) was dissolved in DMSO to make a 10 mM stock solution, and a final concentration of 5 μM was used. Cells were subsequently imaged 24–36 h after the addition of DZNep.

## Live-cell fluorescence microscopy

Imaging of confined cell migration was conducted using Zeiss LSM 980 laser-scanning confocal microscopes, equipped with a Plan-Apochromat 63 ×/1.4 oil immersion objective. Live-cell samples were imaged in a temperature-controlled stage and environment chamber at 5% $CO_2$ and 37 °C. The excitation for mGFP and EYFP, mCherry, and miRFP670 was achieved with lasers at wavelengths 488, 561, and 639 nm, with the transmitter channel using the 639 nm laser. The image acquisition was automated through ZenBlue (Zeiss) software, scanning a large tile region with time-lapse intervals between 5 and 10 min, for a minimum of 12–14 hours. At each time interval, images were acquired in the sequential order from fluorescence to transmitted channel. Image focus was monitored throughout the movie using Zeiss Definite Focus, repeating at every tile or every tile region, with dynamic stabilization every 120 seconds.

For imaging fusion and fission of nucleoli and nuclear speckles, z-stacks at 10 μm total height at a frequency of 3 min were taken on a custom-built spinning disk confocal microscope comprising a Nikon 20x air objective, a Yokogawa CSU-W1 Confocal Scanner Unit, and an Andor DU-897 electron-multiplying charge-coupled device camera mounted on a Nikon Eclipse Ti body, with an Okolab cage incubator at 37 °C and 5% $CO_2$. For light activation experiments of Corelets, cells were plated on 96-well glass-bottom plates, and image acquisition was done with a Nikon Plan Apo VC 100 ×/1.4 oil immersion objective on the same custom-built spinning disk setup. 488, 561, and 640 nm lasers were used to image mGFP, mCherry, and miRFP670 constructs, respectively, on both microscopes.

For PIV mapping of chromatin flow, H2B-mGFP movies of 30 sec to 1 min stream are recorded using a Yokogawa CSU W1 spinning disk confocal microscope with a Nikon Apo TIRF 60 ×/1.49 oil immersion

objective, temperature-controlled at 37 °C and 0.5% $CO_2$. Movie streams are taken with a 488 nm laser at an exposure time of 200 ms or 250 ms done in Ultra-Quiet mode to obtain high temporal resolution. No binning was used; the pixel size for the 60 × objective was 0.108 μm.

### Microfabrication of microfluidic device

The microfluidic devices were designed and manufactured using existing protocols developed in the Lammerding lab[40,43]. Photomask was made using published AutoCAD files with a Heidelberg Mask Writer DWL66 +. The device consists of two layers: the 5-μm tall migration area with PDMS pillars 2 μm apart, bounded by two taller regions for cell attachment with a height of 250 μm. The first layer containing the constrictions was made from a 5 μm thick layer of SU-8 2005 (Kayaku Advanced Materials Inc). Briefly, a silicon wafer was spin-coated with SU-8 2005 according to the manufacturer's protocol. Subsequently, the wafer was baked, covered with a photo mask described above, exposed to UV light, and developed to wash away unexposed photoresist. The wafer was then spin-coated with a 250 μm thick SU-8 100 (Kayaku Advanced Materials Inc) according to the manufacturer's protocol, baked, aligned to the photomask used for the first layer using Suss MA6 Mask Aligner, exposed to UV light, and developed. Subsequently, polydimethylsiloxane (PDMS) and cross-linker (10:1) (Dow Corning, Sylgard 184) were poured onto the wafer, degassed under vacuum, and cured by baking at 65 °C overnight. The PDMS mold was peeled off and cut into individual devices, and holes for cell inlets and media reservoirs were created using 5 mm or 1 mm biopsy punches. Microfluidic devices and glass-bottom 6-well plates were plasma-treated for 1 min using a PE-25 Venus plasma cleaner and immediately assembled by gently pressing the PDMS device onto the 6-well plate for 1 min. Adhesion was improved by incubating the assembled device on a 95 °C hot plate for 1 minute.

### Confined migration assay

The confined migration assay protocol is adapted from the existing protocol[43,73]. MDA-MB-231 cells expressing fluorescently tagged proteins of interest were grown in a 6-well plate. The microfluidic device was covalently bonded to a glass-bottom 6-well plate. To prepare the device for cell seeding, rinse the device with 70% ethanol and then sterile nuclease-free water by pipetting into the inlets. Thoroughly wash a few times with water and inspect the device under brightfield to ensure complete water coverage over the entire PDMS area. After the initial wash, wash the device 3 times with 1x PBS by adding 80 μL to each reservoir, with 5 min incubation at room temperature in between. Ensure the entire device is connected and in contact with PBS by sucking on one reservoir and watching the liquid level going down on the reservoir on the other side. To coat the surface with ECM protein, fibronectin (Sigma #F1141-5MG) is diluted 1:4 (v:v) in 1x PBS and added to the cell seeding inlet. After 15 min incubation at room temperature, the device was rinsed with 1x DMEM 3 times with 5 min incubation in between. MDA-MB-231 cells were trypsinized, counted, and resuspended in 1x DMEM at a density of $5 \times 10^6$ - $8 \times 10^6$ cells per mL. Roughly 80 μL of 1x DMEM and FBS-containing media was added to the reservoirs, and 6 μL of cell resuspension was added to cell inlets. Nuclease-free water was added to other empty wells of the device-containing 6-well plate to prevent media evaporation in the device. Cells were incubated at 37 °C and 5% $CO_2$ for a minimum of 6 h to allow cell attachment, before imaging experiments.

### Fluorescence recovery after photobleaching

MDA-MB-231 cells expressing ferritin core were first seeded to a microfluidic device. FRAP experiments were done on cells undergoing confined migration halfway through the 2 μm constriction ($P = 0.5$). Regions in the advancing half and the trailing half of the same cell were then bleached in 500 nm circular ROIs with the 488 nm laser at high power for 5 frames, with a reference ROI without bleaching in a

different cell. Fluorescence recovery was monitored while imaging the mGFP channel for 60 seconds. Fluorescence intensity was background subtracted and standardized based on the non-bleached ROI control for FRAP-independent bleaching. Fluorescence intensity was compared to the initial image for generating plots. Mobile fraction is measured by fitting the average fluorescence recovery curve using the following equation: $y = b*(1 - e^{-k(t-t_0)})$ where $b$ is the mobile fraction and $t_0$ is the frame of bleaching.

### Quantitative image analysis

**Global and local chromatin intensity analysis.** Chromatin mean intensity was quantified by measuring H2B-mGFP pixel intensities across the entire nucleus, using custom MATLAB scripts. To measure perinucleolar heterochromatin intensity in cells co-expressing NPM1-mCherry and H2B-mGFP, binary masks of the nucleolar rims were segmented based on dilating the NPM1-mCherry binary mask by 6 pixels (0.5 μm). The mean intensity of H2B-mGFP in the binary mask of nucleolar rims was used as a measurement of perinucleolar chromatin density.

**Pairwise mean-square displacement.** GEMs tracking experiments were done in cells that are halfway through the 2 μm constrictions. Images of the mGFP channel were taken every 10 ms for 10 s with a 256 × 256-pixel region of interest using the 488 nm laser on a CSU-W1 spinning disk confocal. Movies were then registered using StackReg to remove the bulk motion of the nucleus. Subsequently, trackMate v7.11.1 in ImageJ was used to identify the tracks of GEMs in each movie. Subpixel tracking was performed using a Laplacian of Gaussian filter-based detector and a blob diameter of 500 nm with a threshold of 30 or adjusted to suit individual samples. Trajectories were then constructed using the simple linear assignment problem (LAP) tracking with max linking and gap-closing distances of 500 nm and no frame gap accepted. Coordinates at each frame were parsed into MATLAB, and only pairs of GEMs co-existing for more than 400 frames were accepted. Corelet tracking experiments were done in cells that are halfway through the 2 μm constrictions. Images of the mGFP-core channel were taken every 60 ms for 2 min with a 256 × 256-pixel region of interest, following 3 min of initial activation using the 488 nm laser on a CSU-W1 spinning disk confocal as described previously[35]. Movies were then registered using StackReg to remove the bulk motion of the nucleus. Subsequently, TrackMate v7.11.1 in ImageJ was used to identify the tracks of Corelet droplets in each movie. Subpixel tracking was performed using a Laplacian of Gaussian filter-based detector and a blob diameter of 500 nm with a threshold of 30 or adjusted to suit individual samples. Trajectories were then constructed using the simple linear assignment problem (LAP) tracking with max linking and gap-closing distances of 500 nm and no frame gap accepted. Coordinates at each frame were parsed into MATLAB, and only pairs of Corelets co-existing for more than 350 frames were accepted. The mean-square displacement of distance between each co-existing pair was calculated with custom MATLAB scripts as $<d^2> = <(x_i(t) - x_j(t))^2>_{t, i \neq j}$ at each lag time and for all pairs $(i, j)$, where $x_i(t)$ is the x coordinate of a Corelet droplet determined by TrackMate.

**Particle imaging velocimetry.** Bulk chromatin in live cells was fluorescently labeled with H2B-mGFP. Movies of 1-min length were recorded while cells were positioned right in the middle of the constrictions, at a frame rate of 200 ms, which is several orders of magnitude faster than the cell migration time scale. The fast movie acquisition allows us to accurately map the chromatin displacement field over the entire nucleus with negligible contribution from the bulk motion of the nucleus. Chromatin flow field and strain were measured by tracking H2B-mGFP in cells that are halfway through the 2 μm constrictions, using the aforementioned CSU W1 spinning disk confocal microscope.

Image acquisition and analysis were done following previously published protocol[45]. H2B-mGFP movies were taken at a frame rate of 200 ms or 250 ms per frame for 30 sec or 1 minute. Cells are included in the PIV analysis based on a similar nucleus size for pre-normalization. Movies were then bleach-corrected by exponential fitting and analyzed using a publicly available package, PIVlab (version 2.61), in MATLAB[74]. Binary masks of the nucleus were generated by Otsu thresholding of the H2B-mGFP movie in Fiji, and imported in PIVlab to filter background signals outside of the nucleus. For PIVlab parameter settings, Fourier transforms correlation with multiple passes (Pass 1: 30 pixel, 15-pixel overlap; Pass 2: 20 pixel, 10-pixel overlap; Pass 3: 10 pixel, 5-pixel overlap) was used to identify the correlation of H2B signal within the window of interrogation, between adjacent frames, to assign a vector for local H2B displacement. FFT analysis was performed for the entire movie. Post-processing was done to filter out displacement vectors with a correlation coefficient below 0.3, and missing vectors were spatially interpolated. The velocity of the chromatin displacement map was calculated using custom MATLAB scripts.

**Nucleus size measurement of volumetric compression via sorbitol.** To quantify the size of the nucleus, we take z-stack images of 22.5 μm with a step size of 300 nm. The individual cell nucleus is then cropped from the z-stack images. The volume of each cell nucleus is determined by thresholding individual nuclei using the Otsu method with a custom-written script in MATLAB.

**Calibration of protein concentration.** The mapping of fluorescence intensity to absolute concentration was done using a U2OS cell line expressing mGFP-P2A-mCherry, with an equimolar amount of intracellular mGFP to mCherry due to the autocatalytic P2A linker. The measurements were performed using Zeiss LSM 980 laser-scanning confocal microscopes, equipped with a C-Apochromat 40x/1.2 W autocorr FCS M27 objective. All measurements and data analysis were performed using the ZenBlue Dynamic Profiler Software (Zeiss) with an AiryScan detector[75]. First, a reference image of the cells is taken at 488 nm, and regions of interest in the cell nuclei are chosen to be measured. Subsequently, data for the concentration of proteins were obtained using 10 s FCS measurement time. mGFP intracellular concentrations (in nM) were measured by fitting the autocorrelation curve to a 1-component 3D diffusion model using the Dynamic Profiler (Zeiss) software. mCherry intracellular concentration was converted by determining the mCherry to GFP fluorescence ratio. Then, further fluorescence calibration between the Zeiss LSM 980 laser-scanning confocal microscopes and the spinning disk confocal microscope (used for Corelet phase mapping) was done by linear fitting of the fluorescence intensities of the mGFP-P2A-mCherry U2OS cells taken on both scopes.

**High-throughput phase mapping.** Cells expressing Corelets components are imaged with a 488 nm laser at an intensity of 3% with an exposure time of 200 ms, and a 561 nm laser at an intensity of 30% with an exposure time of 200 ms to determine the fluorescent intensity of the 'Core' and 'IDR' of the individual cell before activating the phase separation process with blue light (pre-activation). The fluorescent intensity is then converted into physical concentration by using the calibration curve. The cells are subsequently imaged with a 488 nm laser at an intensity of 100% with an exposure time of 1 s to capture the post-activation snapshot. The post-activation images are utilized to classify the cells into a phase-separated (PS) group and a non-phase-separated (non-PS) group, and the Core concentration and core-to-IDR ratio are determined from the corresponding pre-activation snapshots by drawing a rectangular ROI of ~ 30 μm$^2$ inside nuclei but excluding nucleoli, which is then converted into the physical concentration. Phase boundary is determined using an unbiased automated SVM algorithm in MATLAB using polynomial fit as detailed previously[76].

## Statistics and reproducibility

All experiments were performed in $N \geq 3$ independent biological replicates. Sample sizes (n) for all other experiments and analyses are defined in the appropriate figure legends. For confined migration experiments, cells were identified for inclusion in various image analyses based on the criteria of complete migration of the cell nuclei through the constriction. Statistical analysis is done by either $t$ test (two-tailed) for comparison between two groups or one-way ANOVA (two-tailed) for multiple comparisons in more than two groups, with $p$-value detailed in the figure legends.

## Reporting summary

Further information on research design is available in the Nature Portfolio Reporting Summary linked to this article.

## Data availability

All data that support the findings of this study are included in this manuscript and in the Supplementary Information. Source data are provided in this paper.

## Code availability

Codes for image analysis and quantification with this work are publicly available via GitHub at https://github.com/zzhao97/migration.git with associated citable https://doi.org/10.5281/zenodo.14004656.

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

## Acknowledgements

We thank Jan Lammerding for sharing the PDMS microfluidics devices for early experimental development. We thank Britt Adamson for initial advice, Amy R. Strom as well as all other members of the Brangwynne lab for discussion. We thank David W. Sanders for FM5 vectors, Lifei Jiang and Michael Tsai for early assistance with experiments, Yi-Che Chang for comments on the manuscript, Christina DeCoste, Katherine Rittenbach, and Gabriel Palmieri for assistance with FACS sorting experiments, and the Molecular Biology Flow Cytometry Resource Facility which is partially supported by the Rutgers Cancer Institute of New Jersey NCI-CCSG P30CA072720-5921, Roman Akhmechet and Zuzanna Lewicka at the Micro/Nanofabrication Center at the Princeton Materials Institute for training and help for manufacturing PDMS devices, Evangelos Gatzogiannis for microscopy training and assistance. This work was funded by the Princeton Center for Complex Materials, an NSF MRSEC (DMR-2011750); the AFOSR MURI (FA9550-20-1-0241); the St. Jude Research Collaborative on the Biology and Biophysics of RNP granules; and the Howard Hughes Medical Institute.

## Author contributions

J.Z.Z. and C.P.B. conceptualized the study. J.Z.Z. and J.X. performed experiments and formal analyses. J.Z.Z. and C.P.B. wrote the manuscript, and J.Z.Z. made the figures, with contributions from all authors. C.P.B. acquired the funding for this work.

## Competing interests

C.P.B. is a founder of and consultant for Nereid Therapeutics. The remaining authors declare no competing interests.
