## [Transparent Peer Review file · Nature Communications]

Chromatin Compaction During Confined Cell Migration Induces and Reshapes Nuclear Condensates

Corresponding Author: Professor Clifford Brangwynne

Version 0:

Reviewer comments:

Reviewer #1

(Remarks to the Author)

In this manuscript Jessica Z. Zhao et al. study the transient formation of nuclear condensates during migration through constrained channels. They show that nuclear speckles and nucleoli show events of fission, fusion and reversible condensate formation during constricted migration. Further, they observe differential behaviors of chromatin between the front and the back of a constrained migrating nucleus. The front progressively loses the normalized H2B heterogeneity, opposite to the back. Then, they study how condensate-forming proteins show distinct phase behavior in advanced vs. trailing half of the nucleus. The conceptual framework is summed up in a cartoon in figure 6. All these findings are interesting for both the mechanobiology and nuclear condensate fields. The experimental setup is simple and elegant, and the images that are shown are clear. There are, however, several issues that temper the excitement. Specific comments below:

Major Points.

- Although the experimental setups are interesting, the lack of data quantification makes it difficult to make many conclusions. For example, figure 1 has no quantification at all, figure 2 has only one quantified panel. However, many principal conclusions of the paper rely on such data. Authors need to guarantee that all the conclusions through the manuscript are made by properly quantified data and relevant statistics.
- Replicates. The independent biological replicates are claimed to be $N > 2$. I would like to clarify whether this means $N = 3$.
- One of the main conclusions of the paper is that the differential mechanics between the front and the rear of the nucleus is generating, however the only mechanical characterization of differential nuclear mechanics is the use of Corelets as rheological probes and quantify their passive MSD.
- Authors claim to “show that this (de novo condensate formation) arises due to increased chromatin heterogeneity, which gives rise to a shift in the binodal phase boundary”. To test whether changes in chromatin heterogeneity can lead to changes in condensate phase equilibria, authors perform chemical perturbation methods affecting chromatin compaction, and conclude that chromatin compaction affects condensate formation. However, the direct causality of differential mechanics generating de novo condensate formation through chromatin compaction remains indirect. Authors could show this by combining chemical chromatin perturbations with constriction.
- Authors need to show individual traces of data, not only mean and standard deviation (eg. Fig 2D, 5C...)
- “The difference in chromatin behavior in advancing vs. trailing half of the cell nucleus indicates that the mechanical environment of the two compartments are distinct “. It does not directly indicate.
- I strongly suggest that source data and code are available upon publication.

Minor

- In time sequences shown in different figures only passage through constriction is shown. However, quantifications are done with cells before, during and after passing through. Before and after images should be inserted in the time sequences.
- When only one channel is shown (EG. Fig1 panels D, E and F) images are better appreciated in greyscale.
- The control shown in Fig S7 is relevant for data interpretation. I suggest including it in Fig 5.
- “Healthy morphology” is not a scientific term.

Reviewer #2

(Remarks to the Author)

In this paper, the authors embarked on an interesting question: how does nuclear mechanics shape condensate behavior? They find using innovative tools that in a migrating cell going through confinement, the chromatin is more heterogeneous in

the back, and condensates are more likely to form there. I find the paper interesting. However, the authors did not go into depth to analyze their data, making their findings preliminary. The paper needs more substantial work for it to be accepted for publication.

1. The paper described a number of findings without showing them. For example, on page 3, the authors when describing their microfluidics devices, mentioned that "Throughout the entire migration duration, cells show healthy morphology and expected phenotype for migration, including nucleus deformation, cytoplasmic structure protrusion along the direction of migration, and cell rear-end contraction. Cells are able to undergo mitosis undisturbed after migrating through PDMS pillars in both confined and control regions." The authors should show the data mentioned.
2. The authors should quantify the fission and fusion events of nucleoli and nuclear speckles in Fig. 1D-F, in addition to showing representative timelapse. This gives the readers idea of how rare these events are.
3. Data of peri-nucleolar condensation (Fig. 2C) is confusing (it was also mislabelled to Fig. 2D in the text on page 8). If the conclusion of the paper is local decondensation of chromatin in the back of the migrating cell seeds condensates, why would the authors show the condensation data?
4. This sentence is confusing: "This suggests that... the deformation increases chromatin compaction in the back, with preferential localization of chromatin excluding condensates." What is preferential localization of chromatin excluding condensates?
5. I think the major missing link here is the relation between chromatin heterogeneity and condensate formation. The authors argued that decondensed chromatin is more likely to seed condensate formation. They need to provide evidence that it is really the case, by images that condensates indeed form at decondensed areas.
6. Chromatin heterogeneity also leads to more condensed chromatin areas, and the authors also need to explain why it didn't suppress condensate formation.
7. Fig. 5E: what happens after 0.5 progression? Does the intensity decrease?
8. For siRNA, the authors need to show blots confirming effective knockdown (Extended Data Fig. S5A-B).

Reviewer #3

(Remarks to the Author)

In this study, Zhao et al. used microfluidic channels with a variety of condensate-forming fluorescent constructs to nicely demonstrate that confined migration can induce differential changes of chromatin heterogeneity in the front vs. rear halves of the nucleus, causing condensate formation preferably in the rear. Interestingly, the authors found that chromatin condensation via pharmacological perturbation and confined migration likely leads to locally softer regions in the nucleus, shifting the phase boundary and favoring condensate formation. Moreover, accumulation of condensate-forming proteins in the rear may also aid in condensate formation. Overall, the authors described the discovery of confined migration-induced condensate formation and provided a feasible biophysical model of its mechanism.

While the authors provided exciting data on how chromatin heterogeneity and nuclear condensates behave when MDA-MB-231 cells migrate through narrow (2- μ m wide) constrictions, some key quantifications and controls are missing. By adding these analyses, the story would be much stronger.

Major points:

1. In Fig. 1D-F, the authors made important observations of nucleoli and nuclear speckles, but only qualitatively. How often do they undergo fission/fusion? Is there any preference? The authors stated that "fusion events primarily place in the rear of the migrating cell nucleus", and "nucleoli remain fused even after the cells exit the constrictions", but no quantification was given. The authors can use quantification methods similar to that used in Fig. 3F-J.
2. Some sentences seem to be over-interpretation of the data or misplaced. On page 4, the authors stated that "both nucleoli and speckles localize in close proximity to chromatin on their surfaces, suggesting that chromatin dynamics directly impact their fission and fusion." On page 6, the authors stated that "the data above suggest nuclear condensate reorganization is due to chromatin structural changes during confined cell migration". However, the role of chromatin structure was only directly explored when the authors later tagged H2B with mGFP. The role that chromatin structure plays should be referred to as an hypothesis or assumption at this point.
3. In general, data in Fig. 1, Fig. 3, and Fig. 5 lacks proper comparison with the 15- μ m wide control channels. The authors did a good job comparing results between 2- μ m and 15- μ m wide channels in Fig. 2. While comparing between "before", "during", and "after" in 2- μ m is informative, the same data would benefit greatly if comparison can be made with data from 15- μ m wide control channels. The comparison would ensure that the observed changes are specific to nuclear squeezing, but not migration per se.
4. On page 8, the authors stated that "quantification of average chromatin intensity shows that there is no difference between the advancing and trailing halves" and referred to Fig. S2. However, the referred figure cannot be found in Fig. S2.
5. The authors did not quantify the dynamics of Corelet condensates like the quantifications of 53BP1/BRD4 in Fig. 3F-J. It would be interesting to compare between the two types of constructs.
6. The authors nicely plotted phase diagram from many cells in Fig. 4. However, Corelet cells only have two examples

plotted onto phase diagrams (Fig. S4). What would it look like if many cells of the same progression (P) are plotted onto the phase diagram?

7. On page 13, the sentence “this effect does not solely reflect pre-existing condensates retained in the trailing half, since the partition coefficients...” requires elaboration. The nomenclature (partition coefficients) and calculation (I_{dense}/I_{dilute}) can be confusing.

8. Statistics: student’s t test should not be used in the context of comparisons between multiple conditions, which would under-estimate p values, leading to false statistical significance. Tests like ANOVA should be used instead.

Minor points:

1. Nomenclatures need to be consistent. Different terms of “front/advancing vs. rear/back/trailing” halves of the nucleus were used throughout the manuscript, which might confuse the readers.

2. Multiple typos of “nucleus envelope”, which should be “nuclear envelope”.

3. Does serum level affect condensate formation? Since the authors use serum gradient to induce migration through the microchannels, it might be informative to make sure serum level does not affect the condensate constructs used.

4. When calculating intensities/numbers in nuclei “before” entering constrictions or “after” leaving constrictions, the authors may consider averaging values from a few frames, as values often vary/fluctuate from frame to frame in live imaging.

5. It might be helpful to briefly define “valence⁻¹” and walk through the phase diagram in the main text, so that readers not familiar with phase separation can follow the interpretation of the diagram.

6. “PNH” in Fig. 2C needs to be defined.

7. Fig. 3B, why does “middle” appear only in this specific chart? What is its definition?

8. Fig. 3H and 3F-G should swap, as the current order does not match the order in the manuscript main text.

9. On page 9, “quantifying the droplet area, we find that rear droplets are slightly larger than those in the front” seem to be a redundant part of the sentence.

10. In Fig. 5, statistical significance comparing to $P = 0$ should be calculated.

11. Bleomycin validation of RNF168 knock-down (Fig. S5A) needs to be mentioned in the main text.

12. For the discussion sentence on page 15, “we speculate that rear-end local chromatin compaction...”, the authors should also read and refer to Heo et al. 2022 (Nat Biomed Eng), which made interesting observation of local chromatin organization during chromatin compaction and decompaction.

Version 1:

Reviewer comments:

Reviewer #1

(Remarks to the Author)

Authors have fulfilled most of my requests and the manuscript has considerably improved. However, I am still not entirely happy with all the content. Specific comments below:

· Although I acknowledge the difficulty of the Corelets tracking experiment, publishing and making conclusions with $N=2$ is suboptimal. Authors should redo such experiment and show data based on $N \geq 3$.

· Mechanical characterization of cells is challenging in some particular cases as in the one described in the manuscript. However, limitations of the study need to be clear for the reader. Authors should mention in the main text the limitations of their method to infer mechanics.

As a final remark, the strategy of submitting a manuscript including data without quantification at the first instance is inefficient both for the reviewing process and quality of science.

(Remarks on code availability)

Reviewer #2

(Remarks to the Author)

The paper is much improved now with additional quantifications. I have no further suggestions.

(Remarks on code availability)

Reviewer #3

(Remarks to the Author)

In this study, Zhao et al. used microfluidic channels with a variety of condensate-forming fluorescent constructs to nicely demonstrate that confined migration can induce differential changes of chromatin heterogeneity in the front vs. rear halves of the nucleus, causing condensate formation preferably in the rear. Interestingly, the authors found that chromatin condensation via pharmacological perturbation and confined migration likely leads to locally softer regions in the nucleus, shifting the phase boundary and favoring condensate formation.

Moreover, accumulation of condensate-forming proteins in the rear may also aid in condensate formation. Overall, the authors described the discovery of confined migration-induced condensate formation and provided a feasible biophysical model of its mechanism.

In the revised manuscript, the authors comprehensively addressed my previous comments and suggestions. The authors added a large amount of quantification data, especially data from the control 15- μm wide channels, which greatly increased the robustness and credibility of the results and interpretation.

(Remarks on code availability)

RESPONSE TO REVIEWER COMMENTS

Reviewer #1 (Remarks to the Author):

In this manuscript Jessica Z. Zhao et al. study the transient formation of nuclear condensates during migration through constrained channels. They show that nuclear speckles and nucleoli show events of fission, fusion and reversible condensate formation during constricted migration. Further, they observe differential behaviors of chromatin between the front and the back of a constrained migrating nucleus. The front progressively loses the normalized H2B heterogeneity, opposite to the back. Then, they study how condensate-forming proteins show distinct phase behavior in advanced vs. training half of the nucleus. The conceptual framework is summed up in a cartoon in figure 6. All these findings are interesting for both the mechanobiology and nuclear condensate fields. The experimental setup is simple and elegant, and the images that are shown are clear.

We thank the reviewer for the detailed review and recognizing the strengths of our work linking fields of mechanobiology and nuclear phase separation. We have provided a point-to-point response to the additional comments below.

There are, however, several issues that temper the excitement. Specific comments below:

Major Points

- Although the experimental setups are interesting, the lack of data quantification makes it difficult to make many conclusions. For example, figure 1 has no quantification at all, figure 2 has only one quantified panel. However, many principal conclusions of the paper rely on such data. Authors need to guarantee that all the conclusions through the manuscript are made by properly quantified data and relevant statistics.

We thank the reviewer for their insightful comments regarding the need for data quantification. In response, we have strengthened our conclusions by including quantifications, particularly about enhanced dynamics in the fusion and fission of nucleoli and nuclear speckles. We have included the distribution of fusion and fission events for nucleoli and nuclear speckles in **Fig. 1g** and the change in the rate of fusion and fission in **Fig. 1h-i** with relevant statistics specified in the figure legends. For all subsequent quantifications for single cells shown in **Fig. 2c-d** and **Fig. 5c-h**, we have included data for control cells migrating through 15- μm channels for statistics analysis as well as individual traces in **Supplementary Fig. S10** and **Supplementary Fig. S11**, as well as relevant statistics specified in the figure legends.

- Replicates. The independent biological replicates are claimed to be $N > 2$. I would like to clarify whether this means $N \geq 3$.

We have included information of independent biological replicates N that each have measurements done in multiple cells (n) and all quantification in the source data file, indicated by distinct date of experiment performed, or explicitly labeled with 'Rep1', 'Rep2', 'Rep3', etc. For all experiments except Corelets tracking, we have $N \geq 3$ independent replicates. For the case of Corelet tracking, we have $N = 2$ with a total of $n = 4$ cells across two independent experiments. It is a challenging experiment to scale up since corelets condense upon blue-light activation, which could be harmful to cells undergoing confined migration in overnight time-lapse acquisition to capture more cells.

- One of the main conclusions of the paper is that the differential mechanics between the front and the rear of the nucleus is generating, however the only mechanical characterization of differential nuclear mechanics is the use of Corelets as rheological probes and quantify their passive MSD.

The reviewer is correct that we have utilized detailed characterization of structural features, whose changes we have inferred have a mechanical effect that underlies our observations. Precise mechanical measurements within living cells are notoriously challenging. Nonetheless, to address this point, we have included new results from fluorescence recovery after photobleaching (FRAP) experiment of free ferritin core. We hypothesize that since the trailing end is more heterogeneous than the advancing half, the mobile fraction of core that will be freely diffusing in the interchromatin space will be higher. Indeed, our FRAP results show that the trailing end has a higher recovery time at tens of seconds, a timescale similarly probed by Corelets pairwise MSD (**Fig. 3b**). In addition, the trailing half having a higher mobile fraction suggests that the trailing half is a more heterogeneous material. We have added the experimental result to the **Supplementary Discussion** and **Supplementary Fig. S4**.

- Authors claim to "show that this (de novo condensate formation) arises due to increased chromatin heterogeneity, which gives rise to a shift in the binodal phase boundary". To test whether changes in chromatin heterogeneity can lead to changes in condensate phase equilibria, authors perform chemical perturbation methods affecting chromatin compaction, and conclude that chromatin compaction affects condensate formation. However, the direct causality of differential mechanics generating de novo condensate formation through chromatin compaction remains indirect. Authors could show this by combining chemical chromatin perturbations with constriction.

We appreciate the reviewer for their experimental suggestion. We have subsequently included a new experiment combining chromatin decompaction using DZNep with constriction in Corelet-expressing cells. We performed light activation for cells at $P = 0.5$, and see no statistical significance in the difference in size, explaining that chromatin compaction during constricted migration plays a role in tuning condensate behavior (see **Supplementary Fig. S9**).

- Authors need to show individual traces of data, not only mean and standard deviation (eg. Fig 2D, 5C...)

We thank the reviewer for the suggestion and have added individual traces of data together with the mean and standard deviation in **Fig 2c-d** and **Fig 5c-h** in **Supplementary Fig. S10** and **Supplementary Fig. S11**.

- “The difference in chromatin behavior in advancing vs. trailing half of the cell nucleus indicates that the mechanical environment of the two compartments are distinct”. It does not directly indicate.

We have revised this sentence to clarify this point: “The difference in chromatin behavior in the advancing vs. trailing half of the cell nucleus suggests that the chromatin mechanical environment of the two compartments may be distinct.”

- I strongly suggest that source data and code are available upon publication.

We fully agree with the reviewer and value the importance of data transparency. Upon publication, we will make both the source data following the recommendation of Nature Communications and the code publicly available through GitHub.

Minor

- In time sequences shown in different figures only passage through constriction is shown. However, quantifications are done with cells before, during and after passing through. Before and after images should be inserted in the time sequences.

We define “Before” and “After” as before or after the cell nucleus has made contact with the circular PDMS pillars, and “During” as cells at the Progression = 0.5 time point. The three groups are only used in Fig 3 where we show the transient nature of nuclear condensation seen in nuclear proteins tested. Indeed, we agree with the reviewer that having representative images that correspond to the three groups is helpful. We have inserted stills of these images next to the quantifications in **Fig. 3e-f**.

- When only one channel is shown (EG. Fig1 panels D, E and F) images are better appreciated in greyscale.

We agree with the reviewer’s suggestion and have changed the main fluorescence channel in **Fig. 1d-f** to grayscale.

- The control shown in Fig S7 is relevant for data interpretation. I suggest including it in Fig 5.

We thank the reviewer for the suggestion and have included the eYFP-tag only quantification in **Fig. 5e**.

- “Healthy morphology” is not a scientific term.

To reflect the observation shown in **Fig. 1a**, we have provided representative images of cells undergoing mitosis in the channels. We have revised this sentence to “Throughout the entire migration duration, cells show nuclear deformation in the 2- μm channel but not 15- μm , with visible cytoplasmic structures protruding through the channels (**Fig 1a**, zoom-in). After migrating through PDMS pillars in both confined and control regions, cells are still able to undergo mitosis (**Supplementary Fig. S1**).”

Reviewer #2 (Remarks to the Author):

In this paper, the authors embarked on an interesting question: how does nuclear mechanics shape condensate behavior? They find using innovative tools that in a migrating cell going through confinement, the chromatin is more heterogeneous in the back, and condensates are more likely to form there. I find the paper interesting. However, the authors did not go into depth to analyze their data, making their findings preliminary. The paper needs more substantial work for it to be accepted for publication.

We thank the reviewer for their thoughtful feedback and appreciation of our question of interest. We acknowledge that while our current analysis provides a solid foundation, further in-depth exploration of the data would enhance the robustness of our findings. To that end, we have included additional quantifications, especially the nucleoli and nuclear speckles dynamics, along with appropriate statistical analysis across all figures. We believe these revisions will strengthen the paper and provide a more comprehensive understanding of the relationship between nuclear mechanics and condensate behavior. We have provided a point-to-point response below.

1. The paper described a number of findings without showing them. For example, on page 3, the authors when describing their microfluidics devices, mentioned that “Throughout the entire migration duration, cells show healthy morphology and expected phenotype for migration, including nucleus deformation, cytoplasmic structure protrusion along the direction of migration, and cell rear-end contraction. Cells are able to undergo mitosis undisturbed after migrating through PDMS pillars in both confined and control regions.” The authors should show the data mentioned.

We thank the reviewer for pointing out the need to provide visual evidence of all the observations described. To reflect the observation shown in **Fig. 1a**, we have provided representative images of cells undergoing mitosis in the channels. We have revised this sentence to “Throughout the entire migration duration, cells show nuclear deformation in the 2- μm channel but not 15- μm , with visible cytoplasmic structures protruding through the channels (**Fig 1a**, zoom-in). After migrating through PDMS pillars in both confined and control regions, cells are still able to undergo mitosis (**Supplementary Fig. S1**).”

2. The authors should quantify the fission and fusion events of nucleoli and nuclear speckles in Fig. 1D-F, in addition to showing representative timelapse. This gives the readers idea of how rare these events are.

We thank the reviewer for raising this point and we have subsequently quantified the enhanced dynamics in fusion and fission of nucleoli and nuclear speckles (with three independent experiments). We have included the distribution of fusion and fission events for nucleoli and nuclear speckles in **Fig. 1g**, and the change in the rate of fusion and fission in **Fig. 1h-i** with relevant statistics specified in the figure legends.

3. Data of peri-nucleolar condensation (Fig. 2C) is confusing (it was also mislabelled to Fig. 2D in the text on page 8). If the conclusion of the paper is local decondensation of chromatin in the back of the migrating cell seeds condensates, why would the authors show the condensation data?

We thank the reviewer for pointing this out and we have corrected the labeling typo. Indeed our conclusion is that local decondensation of regions with low chromatin intensity seeds condensates. However, the nearby chromatin region also undergoes compaction, leading to an overall increase in heterogeneity (see **Fig. 2d**), while the average chromatin intensity is similar between the advancing and trailing halves (see Supplementary **Fig. S2**). We have clarified this point in the manuscript: “Combining this result with the constant overall H2B average intensity (**Supplementary Fig. S2**), this suggests that while the advancing and trailing halves have a similar amount of chromatin during confined migration, the nuclear deformation increases chromatin compaction in the trailing end.”

4. This sentence is confusing: “This suggests that..., the deformation increases chromatin compaction in the back, with preferential localization of chromatin excluding condensates.” What is preferential localization of chromatin excluding condensates?

We apologize for the confusion that arose from the fact that the concept of chromatin-excluding condensates had not been brought up until later in **Fig. 3**. In **Fig. 3e-f**, we have included the zoom-in of condensates localizing at the chromatin-poor region, and cited previous literature investigating chromatin-excluding condensates (Lee et al *Nature Physics* 2022, Shin et al *Cell* 2018).

5. I think the major missing link here is the relation between chromatin heterogeneity and condensate formation. The authors argued that decondensed chromatin is more likely to seed condensate formation. They need to provide evidence that it is really the case, by images that condensates indeed form at decondensed areas.

We again thank the referee for pointing out gaps in the logic of our previous explanation. Several constructs tested in our manuscript that show *de novo* condensation behavior have been characterized extensively in previous publications from our lab to have nucleation

propensity at chromatin-poor region (Corelet: Lee et al, *Nature Physics* 2021, BRD4: Shin & Chang et al, *Cell* 2018). To fully address the reviewer's concern, in **Fig 3e-f**, we have included merged images of H2B-mGFP with BRD4-miRFP (and separately miRFP-53BP1) including zoom-in at the condensation region to highlight the local low chromatin density.

6. Chromatin heterogeneity also leads to more condensed chromatin areas, and the authors also need to explain why it didn't suppress condensate formation.

We thank the reviewer for raising this good point, which we neglected to explain clearly in the initial submission. Here, we are focusing on whether condensates can nucleate within any regions of the chromatin, and find that if chromatin exhibits increased heterogeneity, condensates more readily form. This is because under more uniform chromatin conditions, there are no low-density chromatin regions, corresponding to more mechanically permissible material environments, in which the condensates can form. Under conditions of increasing heterogeneity, such as in the trailing half of migrating nuclei, these mechanically permissive regions are more prevalent (albeit adjacent to *less* mechanically permissive regions, as the reviewer points out). We have attempted to better explain this point in the text.

7. Fig. 5E: what happens after 0.5 progression? Does the intensity decrease?

We have included data beyond $P = 0.5$ in Supplementary Fig. S11. We notice that the intensity peaks at $P = 0.5$ for SRRM1-eYFP construct, but at different later progression time points. Therefore, we picked $P = 0.5$ for all constructs tested for comparison.

8. For siRNA, the authors need to show blots confirming effective knockdown (Supplementary Fig. S5A-B).

We thank the reviewer for the suggestion and have included the western blot results in the **Supplementary Fig. S6a** with three independent replicates.

Reviewer #3 (Remarks to the Author):

In this study, Zhao et al. used microfluidic channels with a variety of condensate-forming fluorescent constructs to nicely demonstrate that confined migration can induce differential changes of chromatin heterogeneity in the front vs. rear halves of the nucleus, causing condensate formation preferably in the rear. Interestingly, the authors found that chromatin condensation via pharmacological perturbation and confined migration likely leads to locally softer regions in the nucleus, shifting the phase boundary and favoring condensate formation. Moreover, accumulation of condensate-forming proteins in the rear may also aid in condensate formation. Overall, the authors described the discovery of confined migration-induced condensate formation and provided a feasible biophysical model of its mechanism.

While the authors provided exciting data on how chromatin heterogeneity and nuclear condensates behave when MDA-MB-231 cells migrate through narrow (2- μm wide) constrictions, some key quantifications and controls are missing. By adding these analyses, the story would be much stronger.

We thank the reviewer for their detailed and constructive feedback. We are pleased that the reviewer found our findings on chromatin heterogeneity and nuclear condensation during confined migration interesting, and we appreciate your acknowledgment of the biophysical model we proposed. We agree that additional quantifications and controls would further strengthen the study, and therefore we have included more quantifications and appropriate statistical analysis in the revised manuscript. We have provided a point-to-point response below.

Major points

1. In Fig. 1D-F, the authors made important observations of nucleoli and nuclear speckles, but only qualitatively. How often do they undergo fission/fusion? Is there any preference? The authors stated that “fusion events primarily place in the rear of the migrating cell nucleus”, and “nucleoli remain fused even after the cells exit the constrictions”, but no quantification was given. The authors can use quantification methods similar to that used in Fig. 3F-J.

We thank the reviewer for raising this point and we have strengthened the conclusion by quantifying the enhanced dynamics in fusion and fission of nucleoli and nuclear speckles (with three independent experiments). We have included the distribution of fusion and fission events for nucleoli and nuclear speckles in **Fig. 1g**, and the change in the rate of fusion and fission in **Fig. 1h-i** with relevant statistics specified in the figure legends.

2. Some sentences seem to be over-interpretation of the data or misplaced. On page 4, the authors stated that “both nucleoli and speckles localize in close proximity to chromatin on their surfaces, suggesting that chromatin dynamics directly impact their fission and fusion.” On page 6, the authors stated that “the data above suggest nuclear condensate reorganization is due to chromatin structural changes during confined cell migration”. However, the role of chromatin structure was only directly explored when the authors later tagged H2B with mGFP. The role that chromatin structure plays should be referred to as an hypothesis or assumption at this point.

We thank the reviewer for the careful reading of our manuscript. We agree that the role of chromatin mechanics influencing nuclear body dynamics is a leading hypothesis, although it is supported by an extensive set of experiments undertaken here. But we have revised these sentences in the main text on page 4 and page 6 accordingly.

3. In general, data in Fig. 1, Fig. 3, and Fig. 5 lack proper comparison with the 15- μm wide control channels. The authors did a good job comparing results between 2- μm and 15- μm wide channels in Fig. 2. While comparing between “before”, “during”, and “after” in 2- μm is

informative, the same data would benefit greatly if comparison can be made with data from 15- μ m wide control channels. The comparison would ensure that the observed changes are specific to nuclear squeezing, but not migration per se.

We thank the reviewer for their suggestion, and have included control group data for the quantification of nucleoli and nuclear speckle dynamics (**Fig. 1g-i**), perinucleolar heterochromatin compaction (**Fig. 2c**). We have also included data of 53BP1- and BRD4-expressing cells migrating in 15- μ m channels in **Fig. 3g-j**. Here the three categories are defined as 'before', 'during' and 'after' cells having made contact with 15- μ m wide PDMS pillars. We also added data for 15- μ m control groups for all eYFP-tagged proteins tested in **Fig. 5c-h** and performed the statistical analysis at each progression bin comparing 15- μ m vs 2- μ m groups.

4. On page 8, the authors stated that “quantification of average chromatin intensity shows that there is no difference between the advancing and trailing halves” and referred to Fig. S2. However, the referred figure cannot be found in Fig. S2.

We thank the reviewer for pointing this out and attention to detail, and we apologize for the oversight. We have corrected the referred supplementary figure to **Supplementary Fig. S2**.

5. The authors did not quantify the dynamics of Corelet condensates like the quantifications of 53BP1/BRD4 in Fig. 3F-J. It would be interesting to compare between the two types of constructs.

While it's certainly an interesting comparison, this is a technically challenging experiment. Corelets condense upon blue-light activation, which could be harmful to cells undergoing confined migration in overnight time-lapse acquisition. To support our hypothesis of differential chromatin heterogeneity in the advancing vs trailing region of the nucleus, we used a pulse of blue light (3-min) to ensure cell viability. Blue light activation only at the halfway point where progression = 0.5 ensures a fair comparison to other constructs 53BP1 and BRD4 at the same progression time point.

6. The authors nicely plotted phase diagram from many cells in Fig. 4. However, Corelet cells only have two examples plotted onto phase diagrams (Fig. S4). What would it look like if many cells of the same progression (P) are plotted onto the phase diagram?

We appreciate the reviewer for recognizing our effort in mapping phase diagrams in Fig. 4. However, in **Fig. S4**, we are showcasing cells that only exhibit *de novo* condensate in the trailing half, when cells are at Progression = 0.5. These cells are expressing Corelets components at a level close to the binodal line, allowing them to exhibit different phase behavior in the trailing vs advancing half in the same cell. Related to the previous point 5, mapping the phase diagram at the same progression time point is also technically challenging, which would require having cells close to the binodal line to distinguish phase-separated vs non-phase-separated cells.

7. On page 13, the sentence “this effect does not solely reflect pre-existing condensates retained in the trailing half, since the partition coefficients...” requires elaboration. The nomenclature (partition coefficients) and calculation ($I_{\text{dense}}/I_{\text{dilute}}$) can be confusing.

We agree that the nomenclature can be further elaborated for a wider readership. We have added a few sentences to clarify this point in the main text: “To connect to our earlier hypothesis that chromatin heterogeneity modulates the phase boundary shift, we note that a shift of the binodal line will result in a lower concentration of the dilute phase, and potentially a higher concentration of the dense phase (**Fig. 5i**). This phase boundary shift manifests as an increase in the partition coefficient, which is the ratio of the protein concentration inside to outside the condensate.”

8. Statistics: student’s t test should not be used in the context of comparisons between multiple conditions, which would under-estimate p values, leading to false statistical significance. Tests like ANOVA should be used instead.

We thank the reviewer for their suggestion and have subsequently performed relevant statistics using one-way ANOVA for multiple groups comparison and paired t-test for comparing advancing vs trailing half of the same cell, specified in each figure legend.

Minor points

1. Nomenclatures need to be consistent. Different terms of “front/advancing vs. rear/back/trailing” halves of the nucleus were used throughout the manuscript, which might confuse the readers.

We thank the reviewer for the careful reading and we have corrected the terms “advancing vs trailing” to be consistent throughout the manuscript.

2. Multiple typos of “nucleus envelope”, which should be “nuclear envelope”.

We thank the reviewer for pointing out the typos and have corrected all to “nuclear envelope”.

3. Does serum level affect condensate formation? Since the authors use serum gradient to induce migration through the microchannels, it might be informative to make sure serum level does not affect the condensate constructs used.

We appreciate the reviewer’s suggestion. To address this concern, we performed an additional experiment where we tested condensate formation at 10% FBS and no FBS using MDA-MB-231 cells plated on glass-bottom plates. We observed no significant differences in the number of condensates. These findings indicate that serum levels do not impact the formation or stability of the condensate constructs used in our system. We have included these results in the revised manuscript (see **Supplementary Fig. S7**).

4. When calculating intensities/numbers in nuclei “before” entering constrictions or “after” leaving constrictions, the authors may consider averaging values from a few frames, as values often vary/fluctuate from frame to frame in live imaging.

We appreciate the reviewer’s suggestion to ensure the robustness of data collection. Given the randomness of the rare confined migration events in a single overnight movie, it is essential to record each event with the largest field of view (with stitching) while keeping the high spatial resolution required to map chromatin heterogeneity. To address this point, for capturing low spatial resolution events such as fusion and fission of nuclear bodies (nucleoli and nuclear speckles), we used a 20x objective with z-stacks, at a faster frame rate (see Methods for more detail) of 3 minutes per multi-FOV z-stack.

5. It might be helpful to briefly define “valence⁻¹” and walk through the phase diagram in the main text, so that readers not familiar with phase separation can follow the interpretation of the diagram.

We thank the reviewer for pointing out the need for clarification. We have revised the corresponding section and added a description: “By taking the ratio of core to IDR-sspB, we can measure the inverse of valence on the y-axis, valence⁻¹, where valence is the propensity for 24-mer cores to participate in multivalent interaction with one another.”

6. “PNH” in Fig. 2C needs to be defined.

We thank the reviewer for their detailed suggestion and have added the definition of “PNH” next to the zoom-in stills in **Fig. 2c**.

7. Fig. 3B, why does “middle” appear only in this specific chart? What is its definition?

Here we expressed Corelets in cells with various expression levels of IDR and core, and globally illuminated the cell with 488-nm light. After 3-min light activation, we were able to observe small condensates even in the 2- μ m constriction region occupied by the chromatin (here defined as “middle”), which were not observable for other endogenous constructs tested.

8. Fig. 3H and 3F-G should swap, as the current order does not match the order in the manuscript main text.

We have corrected the order of Figure 3 panels and the corresponding paragraphs.

9. On page 9, “quantifying the droplet area, we find that rear droplets are slightly larger than those in the front” seem to be a redundant part of the sentence.

We have shortened this sentence for clarity and thank the reviewer for the observation.

10. In Fig. 5, statistical significance compared to $P = 0$ should be calculated.

We agree with the reviewer for proper statistical comparison for the control group ($P = 0$). To verify the trend is significant from the 15- μm control group, which should also reflect the relative intensity at $P = 0$, we have analyzed cells migrating through 15- μm channels and calculated statistical significance between 2- μm channel group and 15- μm channel group of all measurements at each progression bin ($P = 0.1$, $P = 0.2$, ... $P = 0.5$) using a one-way ANOVA test of multiple comparisons. For partition coefficient quantification, we have compared statistical significance to $P = 0$ labeled in figure legend.

11. Bleomycin validation of RNF168 knock-down (Fig. S5A) needs to be mentioned in the main text.

We thank the reviewer for their careful reading and have now included the **Supplementary Fig. S6a-b** in the main text.

12. For the discussion sentence on page 15, “we speculate that rear-end local chromatin compaction...”, the authors should also read and refer to Heo et al. 2022 (Nat Biomed Eng), which made interesting observation of local chromatin organization during chromatin compaction and decompaction.

We thank the reviewer for the suggestion to reference Heo et al. (2022) and their valuable insights into local chromatin organization during compaction and decompaction. We have read the suggested paper and agree that their findings are highly relevant to our discussion. Accordingly, we have added a citation to Heo et al. (2022) in the discussion section and updated the discussion to reflect the connection between our observations and their work.

RESPONSE TO REVIEWER COMMENTS

Review Round 2

Reviewer #1 (Remarks to the Author):

Authors have fulfilled most of my requests and the manuscript has considerably improved. However, I am still not entirely happy with all the content. Specific comments below:

- Although I acknowledge the difficulty of the Corelets tracking experiment, publishing and making conclusions with $N=2$ is suboptimal. Authors should redo such experiment and show data based on $N \geq 3$.
- Mechanical characterization of cells is challenging in some particular cases as in the one described in the manuscript. However, limitations of the study need to be clear for the reader. Authors should mention in the main text the limitations of their method to infer mechanics.

As a final remark, the strategy of submitting a manuscript including data without quantification at the first instance is inefficient both for the reviewing process and quality of science.

We thank the reviewer for their thoughtful feedback and for recognizing the improvements made to the manuscript. Regarding the Corelets tracking experiment: we fully understand the concern about sample size, and we have now repeated the experiment to include an additional replicate, bringing the total to $N=3$; these are indeed challenging experiments to undertake. The revised figure (**Fig. 3b-c**) now reflects this updated data.

Regarding the limitations of our mechanical characterization method: we have added a discussion of limitations of using imaging-based passive microrheology associated with inferring cellular mechanics using our method. The revised text now includes the following sentence: "Our results show that the advancing vs. trailing halves of confined nuclei have different chromatin mechanics.... **However, given the technical challenges of directly measuring the local mechanics of the nucleus within the confined microfluidics channels, we characterized chromatin using passive tracking of nanoprobes (light-induced Corelets and GEMs) and FRAP experiments in the nucleus. These approaches primarily reflect the local subcellular structure of the nucleus, or 'mesh size', which is linked to but not a direct measurement of the local mechanics.** Despite the limitation, our results suggest that in the trailing half, a locally more open chromatin meshwork is more favorable..."

Reviewer #2 (Remarks to the Author):

The paper is much improved now with additional quantifications. I have no further suggestions.

We thank the reviewer for their further reading and appreciate their favorable views of our manuscript.

Reviewer #3 (Remarks to the Author):

In this study, Zhao et al. used microfluidic channels with a variety of condensate-forming fluorescent constructs to nicely demonstrate that confined migration can induce differential changes of chromatin heterogeneity in the front vs. rear halves of the nucleus, causing condensate formation preferably in the rear. Interestingly, the authors found that chromatin condensation via pharmacological perturbation and confined migration likely leads to locally softer regions in the nucleus, shifting the phase boundary and favoring condensate formation. Moreover, accumulation of condensate-forming proteins in the rear may also aid in condensate formation. Overall, the authors described the discovery of confined migration-induced condensate formation and provided a feasible biophysical model of its mechanism.

In the revised manuscript, the authors comprehensively addressed my previous comments and suggestions. The authors added a large amount of quantification data, especially data from the control 15- μm wide channels, which greatly increased the robustness and credibility of the results and interpretation.

We thank the reviewer for their further reading and appreciate their favorable views of our manuscript.